# 3+2+X: What is the most useful depolarisation input for retrieving microphysical properties of non-spherical particles from lidar measurements using the spheroid model of Dubovik et al. (2006)?

Matthias Tesche[1,2], Alexei Kolgotin[3], Moritz Haarig[4], Sharon P. Burton[5], Richard A. Ferrare[5], Chris A. Hostetler[5], and Detlef Müller[1]

[1]School of Physics, Astronomy and Mathematics, University of Hertfordshire, Hatfield, United Kingdom
[2]now at Leipzig Institute for Meteorology (LIM), Leipzig University, Leipzig, Germany
[3]A. M. Prokhorov General Physics Institute, Moscow, Russia
[4]Leibniz Institute for Tropospheric Research (TROPOS), Leipzig, Germany
[5]NASA Langley Research Center, Hampton, USA

*Correspondence to:* Matthias Tesche (matthias.tesche@uni-leipzig.de)

**Abstract.** The typical multiwavelength aerosol lidar data set for inversion of optical to microphysical parameters is composed of three backscatter coefficients ($\beta$) at 355, 532, and 1064 nm and two extinction coefficients ($\alpha$) at 355 and 532 nm. This data combination is referred to as $3\beta+2\alpha$ or 3+2 data set. This set of data is sufficient for retrieving some important microphysical particle parameters if the particles have spherical shape. Here, we investigate the effect of including the particle linear depolarisation ratio ($\delta$) as a third input parameter for the inversion of lidar data. The inversion algorithm is generally not used if measurements show values of $\delta$ that exceed 0.10 at 532 nm, i.e. in the presence of non-spherical particles such as desert dust, volcanic ash, and under special circumstances biomass-burning smoke. We use experimental data collected with instruments that are capable of measuring $\delta$ at all three lidar wavelengths with an inversion routine that applies the spheroidal light-scattering model of *Dubovik et al.* (2006) with a fixed axis-ratio distribution to replicate scattering properties of non-spherical particles. The inversion gives as additional output parameter the fraction of spheroids required to replicate the optical data. This is the first systematic test of the effect of using all theoretically possible combinations of $\delta$ taken at 355, 532, and 1064 nm as input in the lidar data inversion.

We find that depolarisation information at least at one wavelength already provides useful information for the inversion of optical data that have been collected in the presence of non-spherical mineral dust particles. However, any choice of $\delta_\lambda$ will give lower values of the single-scattering albedo than the traditional 3+2 data set. We find that input data sets that include $\delta_{355}$ give a spheroid fraction that closely resembles the dust ratio we obtain from using $\beta_{532}$ and $\delta_{532}$ in a methodology applied in aerosol-type separation. The use of $\delta_{355}$ in data sets of two or three $\delta_\lambda$ reduces the spheroid fraction that is retrieved when using $\delta_{532}$ and $\delta_{1064}$. Use of the latter two parameters without accounting for $\delta_{355}$ generally leads to high spheroid fractions that we consider not trustworthy. The use of three $\delta_\lambda$ instead of two $\delta_\lambda$ including the constraint that one of these is measured at 355 nm does not provide any advantage over using 3+2+$\delta_{355}$ for the observations with varying contributions of mineral dust considered here. However, additional measurements at wavelength different from 355 nm would be desirable for application to a wider range of aerosol scenarios that may include non-spherical smoke particles that can have values of $\delta_{355}$ that are

indistinguishable from those found for mineral dust. We therefore conclude that — depending on measurement capability — the future standard input for inversion of lidar data taken in the presence of mineral dust particles and using the spheroid model of *Dubovik et al.* (2006) might be $3+2+\delta_{355}$ or $3+2+\delta_{355}+\delta_{532}$.

## 1 Introduction

Over the past two decades, the inversion of multiwavelength aerosol lidar measurements for the retrieval of aerosol microphysical properties (*Müller et al.*, 1998, 1999a, b, 2001; *Veselovskii et al.*, 2002; *Ansmann and Müller*, 2005) matured to a stage that allows for automated and unattended data processing (*Müller et al.*, 2014). The methodology uses multiwavelength lidar measurements of aerosol backscatter and extinction coefficients (i.e. the availability of a $3\beta+2\alpha$ input data set, also referred to as 3+2 data set) and the mathematically correct description of light scattering by small particles to solve the ill-posed inverse problem at hand (*Ansmann and Müller*, 2005). Mie theory is used for the mathematical description of light scattering by particles. By definition, this theory cannot be applied to describe light scattering by non-spherical particles. This constraint causes a problem, as aerosol types such as mineral dust or volcanic ash are of non-spherical shape.

The presence of such non-spherical particles in lidar measurements is identified by non-zero values of the particle linear depolarisation ratio ($\delta$, *Gimmestad* 2008). Spherical particles do not depolarize the emitted laser light, and thus, show values of $\delta$ close to zero. depolarisation-ratio measurements with advanced lidars (*Freudenthaler et al.*, 2009) allow for the retrieval of the contribution of non-spherical particles to the measured intensive optical parameters (*Tesche et al.*, 2009b; *Burton et al.*, 2014), and thus allow for comprehensive aerosol-type characterization (*Burton et al.*, 2012; *Groß et al.*, 2013).

A data base for light scattering by spheroids (Dubovik model, *Dubovik et al.* 2006) developed for the inversion of sun-photometer measurements within the framework of the Aerosol Robotic Network (AERONET, https://aeronet.gsfc.nasa.gov/, *Holben et al.* 1998) has been implemented in the lidar data inversion algorithm used here. The first application of the Dubovik model to lidar measurements of mineral dust has been presented by *Veselovskii et al.* (2010), *Di Girolamo et al.* (2012), *Papayannis et al.* (2012), and *Müller et al.* (2013). *Veselovskii et al.* (2010) performed inversions with the spheroid light-scattering data base on the basis of the traditional 3+2 input data set as well as for a 3+2+1 data set that uses $\delta_{532}$ as additional input. The latter parameter can provide information on the contribution of mineral dust to the total aerosol optical properties. From the comparison of the inversion runs with the different input data sets, the authors conclude that using 3+2+1 provides no advantage over the conventional 3+2 input run in which the spheroid fraction is set a priori to 100%. They attribute this insensitivity (with regard to the use of $\delta_{532}$) to the fact that (i) the Dubovik model has not been specifically designed for lidar applications, i.e. the mathematical description of light scattering at 180°, and (ii) that high values of $\delta_{532}$ can only be obtained for values of particle refractive indices that are below values found from atmospheric observations (*Veselovskii et al.*, 2010). *Papayannis et al.* (2012) present results of the inversion of 3+2 data in the presence of mineral dust while *Di Girolamo et al.* (2012) and *Müller et al.* (2013) used 3+2+1 data sets with depolarisation information at 355 nm and 532 nm, respectively. *Veselovskii et al.* (2016) present results of the inversion of lidar data for mineral dust for the case of the conventional 3+2 input (with spheroid fraction set to 100%) and the 3+2+1 input with depolarisation information at 532 nm. The authors conclude that

it is currently not possible to come to a definitive conclusion as to which input data set leads to a more accurate estimation of dust parameters. Instead, they recommend to use the 3+2 input for measurements of pure dust as these inversions provide more realistic estimations of the refractive index of dust particles. Scattering kernels based on Mie theory cannot represent light scattering by non-spherical particles, i.e. particles that lead to increased $\delta_\lambda$ in a lidar measurement. A way to circumvent this problem is to split the optical input according to the information related to spherical and non-spherical particles. The data obtained in that way can subsequently be used to run the inversion considering only spherical scatterers (i.e. Mie kernels) and non-spherical scatterers (i.e. spheroid kernels), respectively. We provide a detailed discussion of this aspect in Section 3.3. However, the aim of this work is to gain insight into the performance of the inversion using Dubovik's model for mixed dust cases as such scenarios have not yet been considered in earlier studies.

On the one hand, the answer to the question on what inversion input provides the most accurate estimate of dust microphysical parameters requires independent measurements of these parameters. An example for such a study is presented by *Müller et al.* (2013). However, the comprehensive data sets required for such an effort can only be obtained in the framework of dedicated and extensive experiments. On the other hand, there has yet been no systematic estimation of the effect of using different depolarisation input for the inversion of lidar data. Today, depolarisation-ratio profiling is most commonly performed at 532 nm. This explains the use of this wavelength in the studies of *Veselovskii et al.* (2010, 2016) and *Müller et al.* (2013). This wavelength is also the only one for which comparisons of the algorithm performance exist with regard to using the 3+2 and the 3+2+1 data set. Setting a future standard on depolarisation-ratio profiling requires us to assess which wavelength provides the best prospects not only for aerosol characterization but also for using the added information as input to inversion runs. Most inversions that use the Dubovik model focused on pure-dust conditions. Values of $\delta_{532}$ were similar to values observed close to dust source regions (*Freudenthaler et al.*, 2009). Such conditions warrant the use of the 3+2 data set with the spheroid fraction set to 100%. It yet needs to be investigated if depolarisation information also allows for the successful retrieval of aerosol microphysical properties in mixed layers of mineral dust and other spherical aerosol types, i.e. aerosol scenarios that are common during observations of long-range transport of mineral dust in the free troposphere. Finally, the latest developments of realizing depolarisation-ratio profiling at 1064 nm or multiple wavelengths (*Burton et al.*, 2015; *Haarig et al.*, 2017a) leads to the question if these new measurement capabilities might also advance the quality of the inversion of lidar measurements in the presence of non-spherical particles.

In this study, we investigate the effect of using $\delta$ at 355, 532, and 1064 nm as additional inversion input to answer the question:

*What is the optimum choice of $\delta_\lambda$ in the inversion of lidar measurements of non-spherical particles using the spheroid model of Dubovik et al. (2006)?*

We address this question by using 3+2+3 multiwavelength lidar measurements taken under both pure and mixed-dust conditions. Specifically, we assume that values of $\delta_\lambda$ are accurate within their respective measurement error and that the findings of our study are primarily related to the light-scattering model used in the inversion calculations. Section 2 presents an overview of the literature on measurements and modelling of spectral particle linear depolarization ratios of mineral dust particles. The

data sources and inversion setup are introduced in Section 3. The results are presented and discussed in Sections 4 and 5, respectively. We close with a summary and our conclusions in Section 6.

## 2 Spectral $\delta$ for mineral dust from measurements and modelling

The particle linear depolarization ratio $\delta$ as measured by depolarization lidar is the ratio of the particle backscatter coefficients measured at planes of polarisation perpendicular and parallel to the plane of polarisation of the emitted laser light (*Gimmestad*, 2008). Like the lidar ratio and the Ångström exponent, it is an intensive parameter that takes on characteristic values for different aerosol types (*Burton et al.*, 2012). It is most sensitive to the shape of the scattering particles, i.e. $\delta = 0$ for spherical scatterers (or those that appear spherical with respect to the considered wavelength) and increases with particle non-sphericity for the particle shapes commonly found for atmospheric aerosols and ice crystals. Reliable measurements of $\delta$ require careful instrument characterisation together with dedicated calibration efforts (*Freudenthaler et al.*, 2009; *Freudenthaler*, 2016).

Light scattering by non-spherical particles such as mineral dust poses a great challenge for applications in atmospheric science as it cannot be described by Mie scattering. The problem has been addressed by introducing a variety of non-spherical model particles (*Kahnert et al.*, 2014) that differ considerably in the capability to properly describe light depolarization (specifically at the 180° backscatter direction crucial for lidar). Spheroids were found to be particularly versatile for this purpose as they allow for addressing different degrees of non-sphericity in mathematically simple expressions. A specific spheroid model is the one introduced by *Dubovik et al.* (2006). It has been developed for the analysis of AERONET sun photometer measurements, and thus, is optimized for passive remote sensing applications. It has been applied in the inversion of lidar data *Veselovskii et al.* (2010) and to retrieve lidar-specific parameters (*Müller et al.*, 2010; *Shin et al.*, 2018) even though it has not been designed to describe light scattering at 180° backscatter direction. Dubovik's spheroidal model features a defined aspect-ratio distribution which separates it from other applications of spheroidal models used for lidar applications (*Wiegner et al.*, 2009; *Gasteiger and Freudenthaler*, 2014).

Figure 1 presents spectral particle linear depolarization ratios obtained from laboratory (*Miffre et al.*, 2016) and field measurements (*Tesche et al.*, 2009a; *Burton et al.*, 2015; *Groß et al.*, 2015; *Haarig et al.*, 2017a) with lidar, the analysis of AERONET observations using the Dubovik model (*Müller et al.*, 2010; *Shin et al.*, 2018), and modelling using spheroids (*Wiegner et al.*, 2009) and other irregularly shaped particles (*Gasteiger et al.*, 2011) used to mimic the light-scattering properties of mineral dust. The purpose of the figure is to provide some context on the range of particle linear depolarization ratios for mineral dust as inferred from measurements and modelling. Field measurements performed in Morocco (*Tesche et al.*, 2009a), at Barbados (*Groß et al.*, 2015; *Haarig et al.*, 2017a) or with research aircraft over the Caribbean and the Chihuahuan Desert (*Burton et al.*, 2015) show values of $\delta_{\rm d}$ in the range from 0.22 to 0.38 at the common lidar wavelengths 355, 532, and 1064 nm. Laboratory measurements of Arizona Test Dust (*Miffre et al.*, 2016) confirm the field measurements at 532 nm and give a value of 0.35 at 355 nm which is about 0.1 larger than what most field measurements show at this wavelength. However, *Miffre et al.* (2016) investigated only two size distributions and did not consider particles with diameters larger than 800 nm for their study. Figure 1 also gives $\delta_\lambda$ as obtained from the inversion of AERONET sun photometer measurements using Dubovik's model

(*Müller et al.*, 2010; *Shin et al.*, 2018). AERONET derives values at wavelengths that differ from the lidar observations, i.e. 440, 670, 870, and 1020 nm. In general, AERONET-derived $\delta$ are closest to the lidar reference at the longest wavelength. The values at 440 nm are lower than the lidar observations at both 355 and 532 nm though a direct comparison is impossible due to the mismatch in the wavelengths of observations. For comparison, Figure 1 gives the results of modelling studies using Mie scattering ($\delta_\lambda = 0$), a spheroid model with aspect ratios and oblate-to-prolate ratios different from the ones used by the Dubovik model (*Wiegner et al.*, 2009), and irregularly shaped particles (*Gasteiger et al.*, 2011). The latter two options are considered here to illustrate the range of modelling results (grey areas in Figure 1) of the light scattering by non-spherical particles with specific focus on lidar application. The comparison shows that the AERONET-derived values using Dubovik's model are on the lower boundary of the values inferred using other particle geometries at wavelengths smaller than 800 nm. The large values observed with lidar can only be reproduced by *Gasteiger et al.* (2011). The characteristics of Dubovik's spheroidal model, though sufficient to reproduce laboratory measurements over a wide range of scattering angles (*Dubovik et al.*, 2006), might hence be a limiting factor when it comes to lidar applications.

Mineral dust and volcanic ash were generally considered to be the aerosol types that show the highest values of $\delta$ at the wavelengths used in aerosol lidars. This would make it easy to detect their occurrence in a lidar measurement. However, recent case studies have shown that biomass-burning smoke can also show values as high as $\delta = 0.20 - 0.25$ at 355 and 532 nm (*Burton et al.*, 2015; *Haarig et al.*, 2018; *Hu et al.*, 2019). Two of the observed cases of highly depolarising smoke particles are also presented in Figure 1 to show the a contrast to the observed $\delta$ spectra for mineral dust. Light-scattering simulations have shown that such values can be reproduced using spheroids of nearly spherical shape (with aspect ratios close to unity, *Bi et al.* 2018; *Gialitaki et al.* 2019) and agglomerates of spheroids (*Mishchenko et al.*, 2016). In the studies of *Bi et al.* (2018) and *Mishchenko et al.* (2016), the model particles need to contain substantial amounts of non- or weakly absorbing material with an imaginary part of the refractive index below 0.01.

## 3  Data and methods

This section provides an overview of the lidar data used in this study, different methodologies to estimate the contribution of non-spherical particles to the measured optical data, as well as a brief description of the inversion procedure.

### 3.1  Lidar data

To date, few lidar instruments have the capability to measure particle linear depolarization ratios at three wavelengths simultaneously and we refer to *Burton et al.* (2015), *Haarig et al.* (2017a), and *Hu et al.* (2019). Here, we use data of the NASA Langley Research Center's High Spectral Resolution Lidar 2 (HSRL-2) that has been operated aboard the NASA B-200 King Air aircraft in the framework of the DISCOVER-AQ project (https://discover-aq.larc.nasa.gov/) and data taken with the Backscatter Extinction lidar-Ratio Temperature Humidity profiling Apparatus (BERTHA) of the Leibniz Institute for Tropospheric Research (TROPOS) during the Saharan Aerosol Long-range Transport and Aerosol-Cloud-Interaction Experiment (SALTRACE, *Weinzierl et al.* 2017).

HSRL-2 is the second-generation airborne HSRL developed at NASA Langley Research Center. It builds on the heritage of the HSRL-1 system (*Hair et al.*, 2008) but operates at the laser wavelengths of 355, 532, and 1064 nm. The 3+2+3 data collected with HSRL-2 allow for a comprehensive characterization of different aerosol types (*Burton et al.*, 2012) and the retrieval of microphysical particle properties (*Müller et al.*, 2014). Further details on the instrument can be found in *Müller et al.* (2014) and *Burton et al.* (2018).

DISCOVER-AQ measurements with HSRL-2 were screened for observations that showed elevated levels of $\delta_{532}$. The observations were identified as dusty mix (*Burton et al.*, 2012) and include flights during DISCOVER-AQ California 2013 (2 cases), DISCOVER-AQ Texas 2013 (4 cases), and DISCOVER-AQ Colorado 2014 (3 cases). An overview of the DISCOVER-AQ measurement days considered here is given in Table 1. The optical input data for the inversion were obtained in the first step by averaging temporally over several minutes of measurements and in the second step by carrying out data averaging over height layers of 150 m.

3+2+3 measurements with TROPOS' BERTHA lidar during SALTRACE are used to assess the performance of the different inversion input data sets in the presence of pure dust conditions. This test under pure dust conditions is needed as such a scenario was not encountered during DISCOVER-AQ.

While BERTHA had been used to characterize the optical properties of pure dust during the Saharan Mineral Dust Experiment (SAMUM, *Tesche et al.* 2009b), the capability of carrying out triple-wavelength $\delta$ measurements with BERTHA has only recently been presented in *Haarig et al.* (2017a). So far, such measurements have been performed to characterize mineral dust (*Haarig et al.*, 2017a), marine aerosols (*Haarig et al.*, 2017b), and biomass-burning smoke (*Haarig et al.*, 2018). An overview of the SALTRACE measurement days considered here is given in Table 1.

## 3.2 Retrieval of dust fraction from optical data

The particle linear depolarization ratio is an intensive aerosol property that can be applied for aerosol classification (*Burton et al.*, 2012; *Groß et al.*, 2013). Because of its sensitivity to particle shape, it can also be used to separate the contributions of spherical and non-spherical particles to the optical parameters measured with aerosol lidar (*Tesche et al.*, 2009b; *Burton et al.*, 2014) or sun photometer (*Shin et al.*, 2019). This approach generally assumes mixtures with a coarse-mode that is composed of mineral dust and a spherical fine-mode. *Tesche et al.* (2009b, 2011b) use measurements of $\delta_{532}$ together with threshold values representative for pure aerosol types to separate the contribution of dust and biomass-burning smoke to the optical properties measured with multiwavelength aerosol Raman lidar at Cape Verde. Their approach assumes an external mixture of two aerosol types. A generalized form of this method that covers a broader variety of aerosol mixtures has been presented by *Burton et al.* (2014) for measurements with HSRL-1. *Mamouri and Ansmann* (2014, 2017) have refined the aerosol-type separation further using a two-step approach that allows for the separation of contributions of coarse dust, fine dust, and a non-dust aerosol type, i.e. marine or continental aerosol.

In principle, these aerosol-type separation techniques can be used to obtain input data sets for the inversion of lidar data that represent the spherical and non-spherical particles in a mixed aerosol plume, respectively. The inversion could then be run with the conventional 3+2 input data set and the spheroid fraction (see Section 3.3) set to either 0% (i.e. Mie kernels)

or 100%. In this study, however, we aim to test how the inversion performs if we use different combinations of additional depolarization-ratio input that allows us to account for the contribution of non-spherical particles to the optical input data. We use the dust ratio, i.e. the ratio of dust-related to total backscatter coefficient at 532 nm, (i) as an estimate of the dust contribution and (ii) for comparing to the spheroid fraction inferred from the inversion. Dust ratios were either taken from

the DISCOVER-AQ data base (these values have been derived according to *Burton et al.* (2014)), or calculated following *Tesche et al.* (2009b) (with $\delta_d = 0.32$ and $\delta_{nd} = 0.01$ to replicate the values from HSRL-2) and *Mamouri and Ansmann* (2014, 2017) ($\delta_{dc} = 0.39$, $\delta_{df} = 0.16$, and $\delta_{nd} = 0.02$ to obtain the contributions of fine and coarse dust). In the discussion of our findings, we will consider the dust ratio for the two-component (*Tesche et al.*, 2009b; *Burton et al.*, 2014) and tree-component (*Mamouri and Ansmann*, 2014, 2017) mixtures as lower and upper limit, respectively, of the likely dust contribution.

## 3.3  Inversion of lidar data

The inversion of multiwavelength lidar data is based on using light-scattering kernels that were computed on the basis of Mie theory (*Ansmann and Müller*, 2005). *Veselovskii et al.* (2010) were the first to investigate the possibility of using spheroid scattering kernels computed for randomly oriented spheroids (*Dubovik et al.*, 2006). This study and those of *Müller et al.* (2013) and *Veselovskii et al.* (2016) added the $\delta$ at 532 nm to the input data. The information provided by $\delta$ allows for retrieving

the spheroid particle fraction as an additional inversion output parameter. For instance, *Müller et al.* (2013) obtained spheroid fractions close to 100% under conditions of pure Saharan dust as identified by $\delta_{532} \geq 0.31$.

Because depolarization-ratio measurements at 532 nm are most common (*Pappalardo et al.*, 2014; *Baars et al.*, 2016), it will be the first choice of the new standard input for the lidar inversion using spheroid kernels (*Veselovskii et al.*, 2010, 2016). In this paper, we investigate if this input is sufficient for retrieving (some of) the microphysical parameters or if improved results

can be obtained by adding depolarization information at 355 and/or 1064 nm (*Gasteiger and Freudenthaler*, 2014).

Inversion calculations have been performed with eight base functions and by varying the minimum and maximum particle radius of the inversion window between 0.075 and 0.450 $\mu$m and 0.1 and 10.0 $\mu$m, respectively. The inversion uses a single refractive index that is independent of particle size and wavelength. The real part of the refractive index was varied between 1.3 and 1.6 with steps of 0.05 while the imaginary part of the refractive index was set to cover a range from 0 to 0.03 in steps

of 0.005. The spheroid fraction was varied between 0% and 100% in steps of 10%. A spheroid fraction of 100% means that calculations are performed using exclusively spheroid kernels (i.e. the Dubovik model) while a value of 0% refers to using Mie kernels. This setup leads to a total of 3675 solutions per inversion run.

For the measurements listed in Table 1, inversion runs have been performed with depolarization input ranging from zero to three wavelengths. We obtain eight runs per measurement height bin. An overview of the various combinations and the name

of each data set is given in Table 2. The current standard input 3+2 data sets do not account for depolarization information (Set I).

Standard inversion outputs are particle number, surface-area, and volume concentration, and effective radius derived from these parameters, complex refractive index, and single-scattering albedo (SSA). The inversion with spheroid kernels also provides us with an estimate of the contribution of spheroids to the values we obtain for each of the parameters. In the inversion

algorithm, the obtained microphysical properties are used to re-calculate the input parameter and to assess the discrepancy between the original input data and the optical data set that are obtained from the retrieved microphysical properties. In the analysis of the inversion calculations, we have averaged those 140 to 200 solutions (median value of 160 for the different input data sets) that revealed the smallest discrepancy to the optical input data. The mean and median discrepancies found from this approach for the different input data sets are shown in Table 2 together with the range of derived values. In general, median discrepancy increased with increasing number of input data from 1 (no depolarization input) over 7-11 (one depolarization input) and 13-17 (two depolarization inputs) to 22 (three depolarization inputs). The error of the respective parameters have been obtained as one standard deviation by averaging over the number of accepted solutions. The absolute errors presented in Table 2 refer to the median of all the error values for a respective input data set.

## 4  Results

We present selected measurement cases that illustrate the effect of the choice of inversion input data sets on the retrieved aerosol microphysical properties. These case studies describe scenarios of varying concentration of non-spherical particles. We then discuss the results for the entire data set outlined in Table 1.

### 4.1  Example: pure dust

A 3+2+3 measurement conducted with BERTHA on 20 June 2014 during SALTRACE, Barbados (*Haarig et al.*, 2017a; *Weinzierl et al.*, 2017) has been chosen. This case represents nearly pure dust conditions, i.e. a situation dominated by non-spherical particles, and has previously been described by *Mamouri and Ansmann* (2017). However, the dust arrived at Barbados after several days of long-range transport. Its bulk properties might have been modified during transport compared to freshly emitted dust. The profiles of $\beta$, $\alpha$, $\delta$, lidar ratio, and Ångström exponents are shown in Figure 2. High values of $\delta$ of about 0.26 at 532 nm and 0.24 at 355 and 1064 nm and wavelength-independent values of $\alpha$ (extinction-related Ångström exponent of zero) and lidar ratios of 40 to 55 sr are indicative of nearly pure dust conditions. Similar values were observed during SAMUM (*Tesche et al.*, 2009b, 2011a). The circles in the plots of the backscatter and extinction coefficients mark the data that were used as input for the inversion, i.e. 11 sets at 11 height levels between 1.0 and 4.0 km height. The mean $\delta_{532}$ for this height range is 0.26. This value results in a dust fraction of 0.85 with regard to the backscatter coefficient at 532 nm when applying the aerosol-type separation method described by *Tesche et al.* (2009b) and using $\delta_{532}$-values of 0.32 and 0.01 for the dust and non-dust part of the external aerosol mixture, respectively. While higher dust fractions would be desirable to properly represent pure-dust conditions (see, e.g. *Freudenthaler et al.* 2009), the general scarcity of suitable measurement data means that this is the "purest" 3+2+3 dust case available to us at the time of this study.

Figure 3 shows the results we obtained from the inversion of the eight depolarization-related variations of input data (see Table 2). We show the results for the effective radius, the 532-nm SSA, the spheroid fraction, the volume concentration, and the complex refractive index. The inversion of all input data sets shows a decrease of effective radius and volume concentration with height while the real part of the refractive index stays fairly constant. Little difference is visible from the inversion results

for these parameters. We obtain a much clearer separation between the inversion results for Set I (the traditional 3+2 data set) and Sets II to VIII (which include depolarization information) for the SSA, the spheroid fraction, and the imaginary part of the complex refractive index. The high values of $\delta_{532}$ lead to a dust fraction above 80% (dashed line in Figure 3c). Unsurprisingly, Set I is the only one that does not result in a very large spheroid fraction. In fact, spheroid fractions were never found to

exceed 40% when using the traditional 3+2 input regardless of the dust content in the mixed pollution plumes. A similar spheroid fraction of on average 35% has previously been reported for the inversion of 3+2 data sets obtained for Saharan dust (*Veselovskii et al.*, 2010). The inverse problem for the 3+2 input data set is strongly under-determined, and thus, leads to untrustworthy spheroid fractions. This is why the inversion of 3+2 data has only been done for pure-dust conditions with the spheroid fraction set to 100% (*Veselovskii et al.*, 2010). The unrealistic values of spheroid particles obtained for Set I coincide

with SSA values of as low as 0.82 and extremely high imaginary parts of the refractive index of 0.015 to 0.018. These values of SSA (imaginary part) are much lower (higher) than the values we obtain for the other sets. SSA (the imaginary part) is slightly lower (higher) for input data that include $\delta_{355}$ (Sets II, V, VI, and VIII), compared to those that do not include depolarization information at 355 nm (Sets III, IV, and VII). Overall, all input data sets that include depolarization information give similar output data for the case of pure dust conditions while the traditional 3+2 data set gives dramatically different results regarding

the absorbing properties of the aerosols, i.e. imaginary part of the refractive index and SSA.

## 4.2   Example: mixed dust

Figure 4 shows a measurement taken with HSRL-2 on 25 September 2013 in the framework of DISCOVER-AQ Texas. The data present the average of eight minutes of measurement time, i.e. between 2057 and 2105 UTC. This measurement case provides more insight on the sensitivity of data products on optical input data that were taken under mixed dust conditions,

i.e. a situation in which mineral dust is mixed with spherical particles and depolarization values are below the ones generally observed for pure dust. The column aerosol load during this measurement was rather low as indicated by an aerosol optical thickness (AOT) of 0.16 at 532 nm (see Table 1).

    The 3+2+3 profiles in Figure 4 show aerosols in a well-mixed layer up to a height of 2.4 km. The mean value of $\delta_{532}$ is 0.099. This number translates to a dust mixing ratio of 0.346 (Table 1, *Burton et al.* (2012)). The strong wavelength dependence

of the backscatter and extinction coefficients suggests the presence of small particles caused by combustion processes. The Houston area is influenced by oil industry and high volume of traffic. The increased values of $\delta_{532}$ are an indicator for the presence of mineral dust. Consequently, dusty mix and urban/pollution were identified as most abundant aerosol types during the measurement (*Burton et al.*, 2012). However, Figure 4 also shows a strong wavelength dependence of the values of $\delta_{\lambda}$, i.e. we find lower (higher) values at 355 nm (1064 nm) compared to 532 nm.

Figure 5 shows the result of the inversion of the optical data represented by the coloured circles in Figure 4. As for the case of pure dust, the volume-concentration profile follows the shape (profile) of the extensive parameters, i.e. backscatter and extinction coefficient. The lowest values of volume concentration are obtained for the case in which $\delta_{532}$ is used as additional information in data inversion. The highest values are found for inversions that make use of the full set of $\delta_{\lambda}$, i.e. the 3+2+3 data set. Taking into consideration the profiles from all eight inversion runs, however, reveals that the choice of depolarization

input seems to have no major effect on particle volume concentration – particularly not on the shape of the profile. In fact, we find comparably small differences of the values of volume concentration for the different input data sets that are defined by a variable number of depolarization information.

In contrast, the use of a different number of depolarization information results in a much stronger spread of the spheroid fraction. If we use no depolarization information we obtain spheroid fractions that vary between 20% and 30% and change erratically from height bin to height bin. The sets III, IV, and VII (i.e. those with $\delta_{532}$, $\delta_{1064}$, and $\delta_{532} + \delta_{1064}$) result in rather high spheroid fractions between 75% and 90%. This result seems to be a clear overestimate as such conditions would refer to the dominance of mineral dust. This predominance is in disagreement with the dust fraction presented in Figure 4. The most plausible spheroid fractions of around 40% combined with strong vertical homogeneity are found for input data sets that contain $\delta_{355}$, i.e. sets II, V, VI, and VIII. These values are also closest to the mean dust ratio of 0.35 that has been determined from the optical data (see Table 1). Those spheroid fractions follow the profile of the dust ratio (dashed line in Figure 5) quite closely.

The separation of the results for different input data can also be seen in the profiles of the 532-nm SSA and the refractive index in Figure 5. Data sets that show higher spheroid fractions also coincide with SSA values that are up to 0.02 higher than values obtained from optical data sets that include information on $\delta_{355}$. This is because the former data sets give higher real parts and lower imaginary parts of the refractive index than the latter. This study targets the comparison of results we obtain from using different combinations of depolarization information as input. For the second example case, we consider the lower values of the spheroid fraction more realistic, and hence, consider data sets that use $\delta_{355}$ as more trustworthy than data sets that do not include depolarization information at 355 nm.

## 4.3   General findings

Figure 6 presents two cases for which the choice of depolarization input has a profound effect on the retrieved spheroid fraction. In the case of 13 July 2014, the steady decrease of $\delta_\lambda$ with height translates to a similar decrease of the spheroid fraction, but only for data sets that include $\delta_{355}$. In fact, this decrease closely follows the decrease of dust fraction with height. As for the previous cases, no variation with height is found when using the traditional 3+2 data set. Sets III, IV, and VII, all of which are lacking depolarization information at 355 nm, do not result in spheroid fractions smaller than 80%. The case of 17 July 2014 is even more striking as – in contrast to the previous examples – $\delta$ is low at all wavelengths and the dust fraction obtained from the optical data is actually zero. Despite this clear pattern of the optical data, the inversion of the different input data sets gives a wide range of spheroid fractions: below 20% for Sets II, V, VI, and VIII; slightly higher values of up to 30% in the lower half of the aerosol layer for Set I; values between 40% and 70% for Sets III and VII; and more than 70% for Set IV. This outcome suggests that using $\delta_{1064}$ does not improve the performance of our particular inversion method using the spheroid Dubovik model. This result is in contrast to the results presented by *Gasteiger and Freudenthaler* (2014). While $\delta_{1064}$ certainly does provide additional information content, this cannot be exploited when combining the Dubovik model with the inversion algorithm used in this study.

A more complete picture of the effect of the choice of $\delta_\lambda$ on the retrieved spheroid fraction is provided in Figure 7. The figure includes all 156 data points obtained from the cases listed in Table 1. The results we obtain from the eight inversion runs is split according to data sets that include, respectively do not include $\delta_{355}$. As in the case of the examples shown before, we consider the retrieved spheroid fraction as the microphysical manifestation of the optically-derived dust ratio (*Tesche et al.*, 2009b). Figure 7 clearly shows that only input data sets that include $\delta_{355}$ lead to any meaningful correlation between dust ratio and spheroid fraction. The parameters of the linear regressions presented in Figure 7 are listed in the bottom half of Table 2. The steepest slope and largest values of the squared correlation coefficients are found for Sets V and VIII, i.e. the sets that either use input values for $\delta_\lambda$ at 355 and 532 nm or all $3\delta_\lambda$, respectively. Figure 7b confirms that (i) spheroid fractions above 40% are impossible to obtain from traditional 3+2 data sets, (ii) the data sets without $\delta_{355}$ give spheroid fractions that are poorly correlated to the obtained dust ratios, and (iii) data sets that include $\delta_{532}$ but not $\delta_{355}$ result in increased spheroid fractions that increase with increasing dust ratio but rarely stay below 40%. We therefore conclude from Figure 7 that in the case of using the spheroid model of *Dubovik et al.* (2006) to describe light scattering by non-spherical particles in the inversion of lidar, $\delta_{355}$ has a regulating effect on the inversion output and that data sets that include $\delta_{355}$ are generally more trustworthy (certainly with respect to the spheroid fraction) than those that do not include $\delta_{355}$.

In the following, we are hence contrasting the results for the volume concentration, the effective radius, and the SSA according to the two sub-sets shown in Figure 7a and b. We want to find out if these parameters differ within and between these two groups. The difference between using 3+2+1 and 3+2+2 input data sets from the two groups (i.e. with and without $\delta_{355}$) is shown in Figure 8 for volume concentration, effective radius, 532-nm SSA, and complex refractive index. The correlation between Sets II and III and between Sets V and VII shows little difference for the effective radius. The use of 3+2+2 input data generally gives larger volume concentration but again little difference is found between the data sets considered in this work. The real part of the refractive index tends to be smaller for the input data sets that include $\delta_{355}$, though this effect diminishes towards values of 1.55. The strongest effect with regard to the choice of input data is found for the imaginary part of the refractive index, and thus, the SSA. Input data that include $\delta_{355}$ tend to give lower values of SSA (Figure 8c, see also Figures 3 and 5) as a result of a general shift towards higher imaginary parts (Figure 8e). Figure 8d shows a similar, though less pronounced tendency for the real part: data sets that include $\delta_{355}$ give values between 1.45 and 1.58 while most values are in the range from 1.50 to 1.58 for data sets that include $\delta_{532}$. of SSA between 0.90 and 0.98 for Sets II and V whereas we find a considerably narrower range from 0.96 to 0.98 for Sets III and VII. The same behaviour is found when the correlation analysis in Figure 8 is expanded to the remaining input data sets (not shown).

The dependence of the retrieved real and imaginary parts of the refractive index on the dust ratio is shown in Figure 9. The upper plot shows that the retrieved real parts cover a large range of values for dust ratios smaller than 60%. In addition, data sets that include $\delta_{355}$ generally giving lower values (3+2+3 gives the lowest values) than data sets that exclude $\delta_{355}$. The range of results narrows for larger dust ratios for all inversion input that include depolarization information. Only the traditional 3+2 data set gives values smaller than 1.50 for dust ratios larger than 60%. A reversed behaviour is found for the imaginary part of the refractive index in Figure 9b. Data sets that include $\delta_{355}$ generally give larger values than those that exclude $\delta_{355}$ with the

traditional 3+2 data set leading to values that are by far the highest. This holds particularly true for dust ratios larger than 60% for which the range of results obtained from all other input data sets narrows in the same way we found for the real part.

## 5   Discussion

We would like to start the discussion by emphasising that the results presented here are specific to the application of the spheroid model of *Dubovik et al.* (2006). This light-scattering model has been developed for application to sun photometer measurements in the framework of AERONET. The Dubovik model marks a considerable advance when compared to treating light scattering by non-spherical particles with Mie theory. However, it considers rather simplified particle shapes, i.e. rotational symmetric spheroids with (i) a defined axis-ratio distribution and (ii) a fixed mixture of oblates and prolates. Because of these constraints, the Dubovik model cannot be regarded as a universal spheroid model. In fact, the specific model setup needs to be considered in comparisons to findings from other spheroid models. In addition, the Dubovik model might not be suitable to reproduce all of the light-scattering properties of non-spherical particles in the atmosphere. Indeed, comparisons to independent in-situ measurements and lidar observations of Saharan dust during SAMUM have revealed discrepancies for the retrieved complex refractive index and single-scattering albedo (*Müller et al.*, 2010). In addition, the intensive lidar parameters lidar ratio and particle linear depolarization ratio, which can be calculated from the inferred scattering matrix, do not agree with coincident measurements at the 355- and 532-nm lidar wavelengths (*Müller et al.*, 2013).

*Shin et al.* (2018) present spectral lidar ratios and particle linear depolarization ratios representative for mineral dust from different source regions. The authors used the AERONET data base of level 2.0 sun photometer inversions. They find the best agreement to lidar observations of both parameters at the longer wavelengths of 870 and 1020 nm (Figure 1). Towards shorter wavelengths, the AERONET-derived values show an increase of the lidar ratio and a decrease of the particle linear depolarization ratio. Both spectral behaviours are not found in lidar measurements of mineral dust (*Freudenthaler et al.*, 2009; *Shin et al.*, 2018). Other models that employ more realistic geometries of non-symmetric non-spherical particles (*Gasteiger et al.*, 2011), account for surface roughness (*Kemppinen et al.*, 2015a) and particle inhomogeneities (*Kemppinen et al.*, 2015b), or employ spheroids with wider ranges of the oblate-to-prolate ratio and the aspect ratio (*Gasteiger and Freudenthaler*, 2014) have been developed either specifically for use in lidar applications or used to investigate lidar-specific parameters. These models suggest improvements in inferring aerosol microphysical properties from lidar data by using measurements of $\delta_{1064}$ in the inversion (*Gasteiger and Freudenthaler*, 2014). However, such alternatives generally lack the flexibility of the Dubovik model when it comes to their implementation for new applications. In addition, there are still enormous challenges involved in testing these alternatives in view of the complexity of particle shapes and the computational resources required for running simulation studies.

The results we obtain from our study are somewhat contradictory to the findings of *Gasteiger and Freudenthaler* (2014) and *Shin et al.* (2018) who attribute the greatest informational value and representativeness to observations of $\delta_{1064}$. This is likely due to the difference in the setup of the different studies. On the one hand, *Gasteiger and Freudenthaler* (2014) applied a spheroid model that was constrained in neither the aspect ratio nor the oblate-to-prolate ratio of the spheroids. On the other

hand, *Shin et al.* (2018) are referring to AERONET data which have been derived for scattering angles smaller than 180° (and extrapolated to the backscatter direction) and represent values for the entire atmospheric column. However, the strong and weak effects of using $\delta_{355}$ and $\delta_{1064}$, respectively, in the inversion of lidar measurements of dust-containing aerosol layers based on the spheroid model of *Dubovik et al.* (2006) also indicate that the model's constraints on the aspect-ratio distribution

and the ratio of oblates to prolates have a strong effect on making full use of the informational content provided at different wavelengths in lidar applications. Furthermore, *Gasteiger et al.* (2011) have discussed the possibility that $\delta$ becomes similar for spheroids and irregularly-shaped particles for larger size parameters. For the larger size parameters equivalent to measurements at 355 nm, it might therefore be that $\delta_{355}$ as obtained for spheroid particles is similar to that of more realistic, irregular-shaped particles. As $\delta_{532}$ and $\delta_{1064}$ refer to smaller size parameters, it would consequently be more likely that spheroids fail to properly

describe light-scattering properties at these wavelengths.

We stress again that the conclusions of this study are valid only for the inversion of lidar data that resorts to describe the light-scattering properties of non-spherical dust particles by means of the spheroid model of *Dubovik et al.* (2006) and that any finding might be strongly related to the weaknesses of this particular light-scattering model (*Müller et al.*, 2010, 2013). Nevertheless, no other model has been applied as widely in the inversion of lidar data (*Veselovskii et al.*, 2010; *Di Girolamo et al.*,

2012; *Papayannis et al.*, 2012; *Müller et al.*, 2013). In addition, there has so far been no systematic investigation of the usefulness of different depolarization input for this particular inversion setup. We therefore believe that this work will contribute to a better understanding of the usefulness and limitations of the model of *Dubovik et al.* (2006) in lidar applications as well as to further emphasize the need for a more general model to describe light scattering by non-spherical particles at very large scattering angles.

Following on the initial work of *Veselovskii et al.* (2010), we have performed the first systematic investigation of the effect of all possible combinations of depolarization-related inversion input at the wavelengths of 355, 532, and 1064 nm on the retrieved aerosol microphysical properties. The aim of this study is to assess the performance of the inversion procedure to lidar measurements conducted in the presence of mixture of spherical and non-spherical particles. So far, inversions on the presence of mineral dust have only been attempted under the assumption of pure dust conditions (*Veselovskii et al.*, 2010;

*Di Girolamo et al.*, 2012) or if the contribution of the non-spherical scatterers had been screened from the optical input data (*Tesche et al.*, 2011a, b). We consider the the retrieved spheroid fraction needed to reproduce the measured optical properties as an indicator for the performance of our inversion setup. Based on this parameter, we conclude that any combination of depolarization ratios that includes $\delta_{355}$ provides a useful addition to the 3+2 data set for the inversion of lidar data that have been collected in the presence of non-spherical particles using the Dubovik model. We emphasize again that this finding refers

to using Dubovik's model with its particular setup of resorting to the use of randomly oriented spheroids for the description of light scattering by non-spherical particles in this inversion algorithm.

Our findings provide insights that go beyond previous studies that investigated the effect of adding depolarization information to the inversion of multiwavelength lidar data:

1. Previous studies that also resort to using the Dubovik model for lidar applications focused exclusively on pure-dust

situations, i.e. values of $\delta_{532}$ of 0.30 (*Veselovskii et al.*, 2010, 2016; *Müller et al.*, 2013). These studies showed that the

depolarization ratio should not be used as input for the inversion of dust particle parameters. Instead, the inversion should be performed with 3+2 input and the spheroid fraction manually set to 100%. Our results show that this conclusion may have been driven by using $\delta_{532}$. Our study shows that $\delta_{532}$ may not be an ideal input parameter.

2. We present the first results of applying the inversion with the Dubovik model to lidar observations of mixtures of spherical and non-spherical particles of varying degree and varying spectral behaviour of the particle linear depolarization ratio. Considering such conditions rather than only pure-dust cases allows for using the retrieved spheroid particle fraction as an additional indicator for the quality of the inversion results.

3. We present the first systematic (though relational) study of the effect of the choice of depolarization input based on actual atmospheric triple-depolarization-ratio measurements. Previous investigations of the effect of depolarization input on the inversion results for which Dubovik's model was used have been restricted to using either $\delta_{355}$ (*Di Girolamo et al.*, 2012) or $\delta_{532}$ (*Veselovskii et al.*, 2010, 2016; *Papayannis et al.*, 2012; *Müller et al.*, 2013), and thus, could relate the findings only to the results of using the conventional 3+2 input data set. The lack of spectral depolarization-ratio measurements under dusty conditions neither allowed for investigating how the choice of input parameters affects the quality of inversion results compared to benchmark data nor test if the choice is ideal.

4. Following the footsteps of AERONET's data processing, microphysical particle properties mark the next logical data product level in the analysis of multiwavelength aerosol lidar data. It is therefore of vital importance to define the minimum information needed for this purpose (i.e. the best choice of input data) as this decision relates directly to the optimum setup for lidar instruments whose measurements can provide this data product. This study represents an important step for determining that information though it is restricted to a very specific model that is used for describing the light-scattering properties of non-spherical particles. The main issue in that regard is weighting the benefits of using instrument setups which are already highly challenging over the added information provided by these measurements. This decision-making is of particular importance in light of future spaceborne lidar missions that will focus on aerosol profiling as well as their airborne demonstrators.

*Veselovskii et al.* (2010, 2016) use the complex refractive index in their argumentation of their preference of rejecting the depolarization input in the inversion. They retrieve real parts of about 1.45 for pure dust, which are comparable to AERONET results. They conclude that imaginary parts obtained from the inversion of 3+2 input data lead to more realistic estimations of this parameter because values below 0.005 (derived from using 3+2+1) are below the findings from in-situ measurements (*Müller et al.*, 2013). Our analysis of the refractive index shows real parts of 1.50 to 1.55 for all combinations of depolarization input for both mixed- as well as pure-dust conditions. This result is more in line with independent measurements of this parameter (*Müller et al.*, 2013). For pure dust conditions we obtain imaginary parts of up to 0.020 from the inversion of 3+2 data sets. All other sets lead to significantly lower values. For mixed-dust cases in which $\delta_{532} < 0.25$, we find significantly larger imaginary parts that show little difference to the results obtained from using the 3+2 input data set.

The inversion assumes a spectrally independent complex refractive index. In contrast, mineral dust is known to show a strong increase of the imaginary part of the refractive index with smaller wavelengths. This issue has been explored by *Veselovskii et al.* (2010) who conclude that (i) the error of the volume concentration is estimated as 17% to 25% depending on the contribution of large particles and (ii) a fixed imaginary part refers to the mean value of the spectrally dependent imaginary part. A detailed investigation of the assumption of spectrally independent refractive indices is beyond the scope of this study.

## 6   Summary and conclusions

We have performed a first systematic relational investigation of the effect of exploiting different combinations of depolarization information as input to the inversion of optical lidar data into aerosol microphysical properties. The inversion is run with spheroid kernels based on the Dubovik model for the description of light scattering by non-spherical particles. In this work, we use 3+2+3 measurements obtained with the NASA LaRC HSRL-2 during DISCOVER-AQ and the TROPOS BERTHA during SALTRACE — two out of just three lidar instruments currently capable of measuring $\delta_\lambda$ simultaneously at three wavelengths.

We have selected eleven observations. Increased values of $\delta_{532}$ can be used as a proxy for the presence of an increased concentration of mineral dust in atmospheric layers. Eight sets of optical data have been created for each of the individual measurements. depolarization input ranged from zero to three wavelengths. We focused on a relational study in view of the challenges connected to (i) using the AERONET light-scattering model of *Dubovik et al.* (2006) that currently provides the best possible output results for sun photometer observations and (ii) the lack of light-scattering models that are proven to work for the special condition of observing non-spherical particles at 180-degree observation angle (lidar configuration). We are comparing the output of the different inversion runs to each other and to the dust ratio obtained from the optical data. In that way we want to identify the most plausible results that can be obtained from our specific inversion setup.

We find that inversion without depolarization information (i.e. the traditional 3+2 data set) cannot lead to spheroid particle fractions larger than 40% even if spheroid kernels, i.e. the spheroid Dubovik model, are used. We also find that the use of depolarization ratios at 532 or 1064 nm in combination with the Dubovik model give unrealistically high spheroid particle fractions. These fractions generally exceed the dust ratio inferred from the measurements of $\beta_{532}$ and $\delta_{532}$ following the procedure described by *Tesche et al.* (2009a). While it needs to be emphasized that the spheroid fraction as inferred from the inversion is an artificial, non-physical parameter, it might be considered as the ratio of the concentration of dust to total particle concentration for this particular study. The most realistic spheroid fraction in relation to the lidar-derived dust ratio is found when using depolarization information at 355 nm.

The choice of depolarization input wavelength was found to have little effect on the retrieval of extensive parameters such as the volume concentration and the effective radius that can be derived from this extensive parameter. The use of depolarization input at any wavelength, i.e. 355 nm or 532 nm or 1064 nm, generally increases the retrieved values of the 532-nm SSA compared to the 3+2 input. This is because the use of depolarization information leads to lower values of the imaginary part compared to the inversion in which the traditional 3+2 data set is used. Consequently SSA increases. We conclude from our

relational investigation that any choice of input data that contains $\delta_{355}$ seems to provide more reasonable results of the spheroid inversion than input data sets without $\delta_{355}$ or any depolarization information at all. However, we do not find a significant advantage of using three $\delta$ over using $\delta$ at fewer wavelengths. This result leads us to conclude that the most suitable input data set for lidar inversion using spheroid kernels according to *Dubovik et al.* (2006) is 3+2+1 in which we use $\delta_{355}$.

We investigated the connection between output from different sets of input data in and inversion in which the light-scattering properties of non-spherical particles are described by the spheroid model of *Dubovik et al.* (2006). Definite conclusions can only be drawn if coincident independent in-situ data were available for the considered cases. An alternative approach to circumvent any reliance on the accuracy of the spheroid fraction would be to separate the optical input data according to the contributions of spherical and non-spherical particles (*Tesche et al.*, 2009a, 2011b) before running the inversion with
spheroid fractions set to zero and unity, respectively. Alternatively, a more universal spheroid model without a fixed oblate-to-prolate ratio or aspect ratio distribution (*Gasteiger and Freudenthaler*, 2014) could be applied in the inversion system. In any case, the use of spheroids for approximating light scattering by non-spherical particles in lidar applications is rather limited (*Müller et al.*, 2010). New models with more realistic particle geometries (*Kahnert et al.*, 2014; *Nousiainen and Kandler*, 2015) will be needed to accurately link microphysical properties to the optical parameters measured with advanced aerosol
lidars (*Gasteiger et al.*, 2011). It is quite possible that such improved light-scattering models will show better capability for extracting the informational content provided by particle linear depolarization ratios at 532 and 1064 nm.

*Data availability.* DISCOVER-AQ data are publicly available from the Science Team at the NASA Atmospheric Science Data Center (ASDC) via doi:10.5067/Aircraft/DISCOVER-AQ/Aerosol-TraceGas. SALTRACE data are available from M. Haarig upon request.

*Author contributions.* MT, DM, and AK had the idea for this study and performed the inversion runs. SPB, RAF and CAH collected the
HSRL-2 data during DISCOVER-AQ. MH collected the BERTHA data during SALTRACE. MT performed the analysis and interpretation of the inversion data and prepared the figures. All authors contributed to the discussion of the findings and the preparation of the manuscript.

*Competing interests.* The authors declare that no competing interests are present.

*Acknowledgements.* This activity is supported by ACTRIS Research Infrastructure (EU H2020-R&I) under grant agreement no. 654109.

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

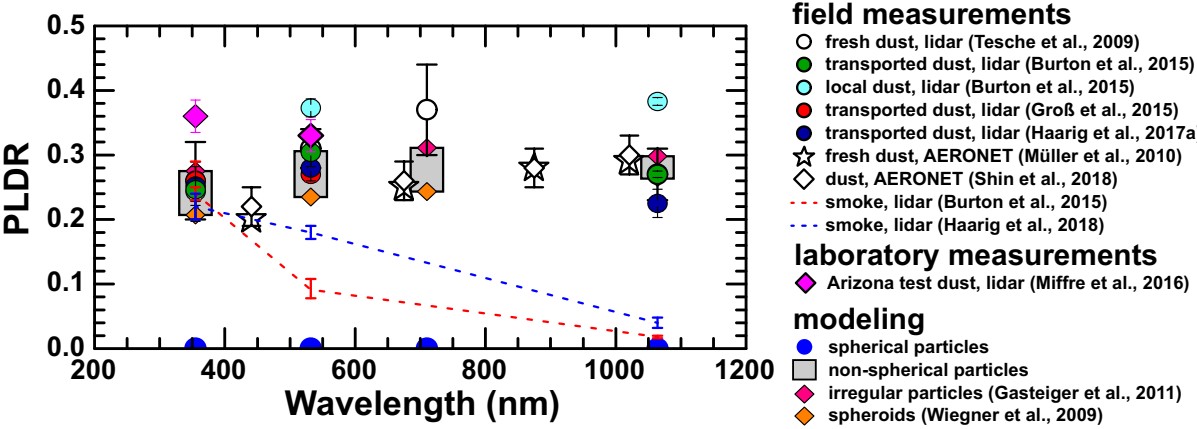

**Figure 1.** Overview of particle linear depolarization ratios for pure mineral dust from field measurements (*Tesche et al.*, 2009a; *Burton et al.*, 2015; *Groß et al.*, 2015; *Haarig et al.*, 2017a), laboratory studies (*Miffre et al.*, 2016), AERONET observations (*Müller et al.*, 2010; *Shin et al.*, 2018), and modelling (*Wiegner et al.*, 2009; *Gasteiger et al.*, 2011). The use of Mie theory (spherical particles) gives values of zero. The range for modelling results is defined by the studies of *Wiegner et al.* (2009) (lower boundary, spheroids) and *Gasteiger et al.* (2011) (upper boundary, mixtures of irregularly shaped particles). The blue and red dashed lines show that field measurements can reveal high depolarization ratios for biomass-burning smoke as well (*Burton et al.*, 2015; *Haarig et al.*, 2018).

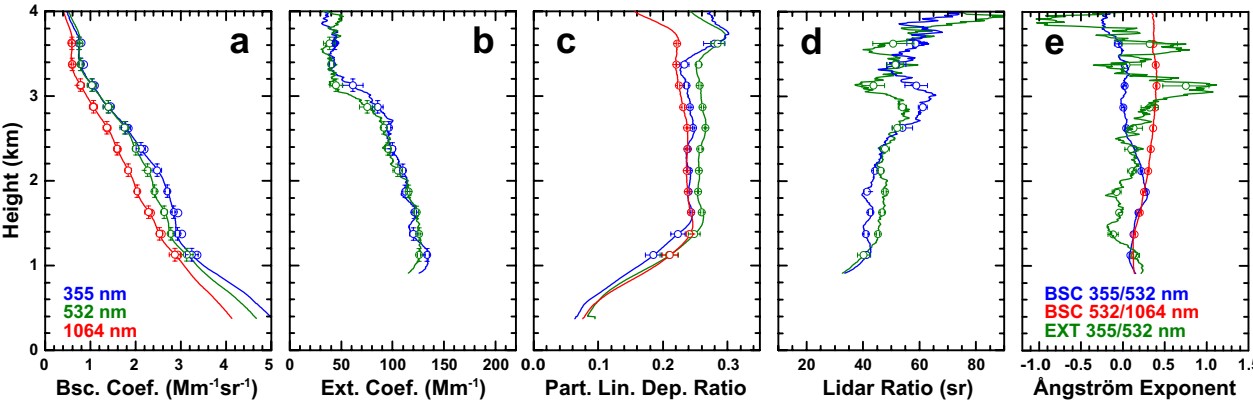

**Figure 2.** BERTHA measurement from 2310 to 0210 UTC on 20-21 June 2014 during SALTRACE in terms of (a) backscatter coefficient, (b) extinction coefficients, (c) particle linear depolarization ratios, (d) lidar ratios, and (e) backsatter- and extinction-related Ångström exponents. Colours mark the corresponding wavelengths or wavelengths pairs. The measurement is representative for pure mineral dust conditions (*Haarig et al.*, 2017a). The coloured circles mark the height averages used to compile the eight variations of input data sets for our data inversion (Table 2). Error bars refer to the standard deviation of the height average. Further details on this measurement are given in Table 1.

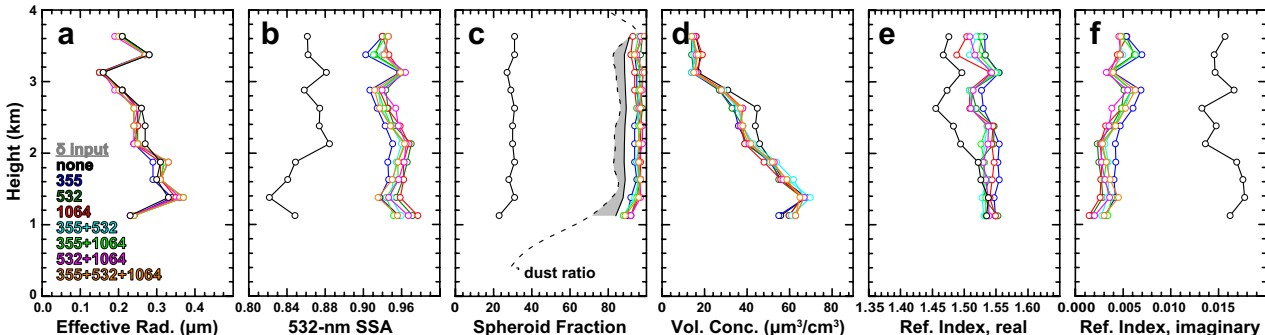

**Figure 3.** Inversion results of (a) effective radius, (b) 532-nm SSA, (c) spheroid fraction, (d) volume concentration, and (e) real and (f) imaginary part of the refractive index for eight inversion runs with varying depolarization-ratio input (colours, see also Table 2) using the input data presented in Figure 2. The dashed and dash-dotted lines in the plot of the non-spherical fraction (c) refers to the contribution of dust to the 532-nm backscatter coefficient that can be obtained according to the procedures described by *Tesche et al.* (2009b) (dust, two-component mixture) and *Mamouri and Ansmann* (2017) (fine and coarse dust, three-component mixture), respectively. The grey area between the two lines marks the likely range of the dust ratio as defined by the two approaches.

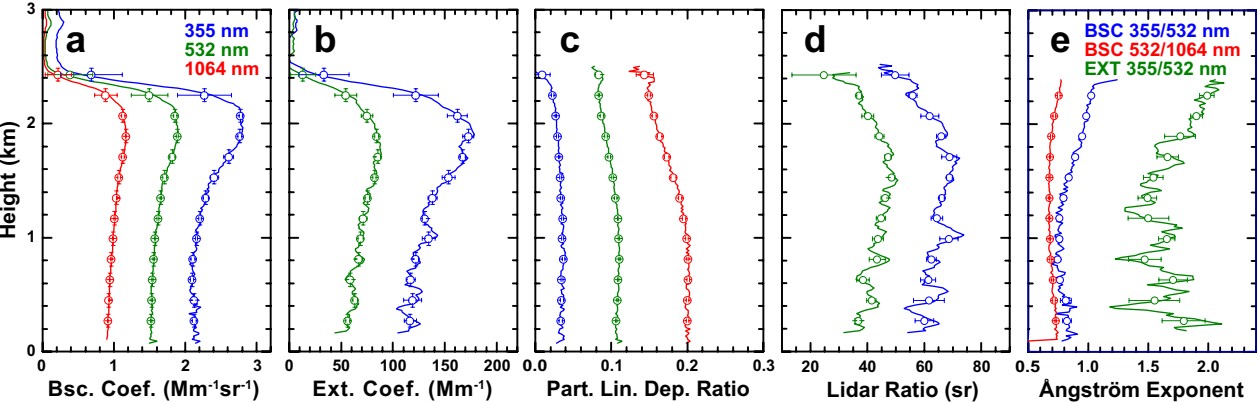

**Figure 4.** Same as Figure 3 but for an HSRL-2 measurement performed during the second DISCOVER-AQ Texas flight on 25 September 2013 in the vicinity of Deer Park (29.670°N, 95.128°W). The coloured circles mark the data points we used to compile the inversion input data sets (Table 2). Details on the time of flight, and dust mixing ratio and aerosol types are given in Table 1.

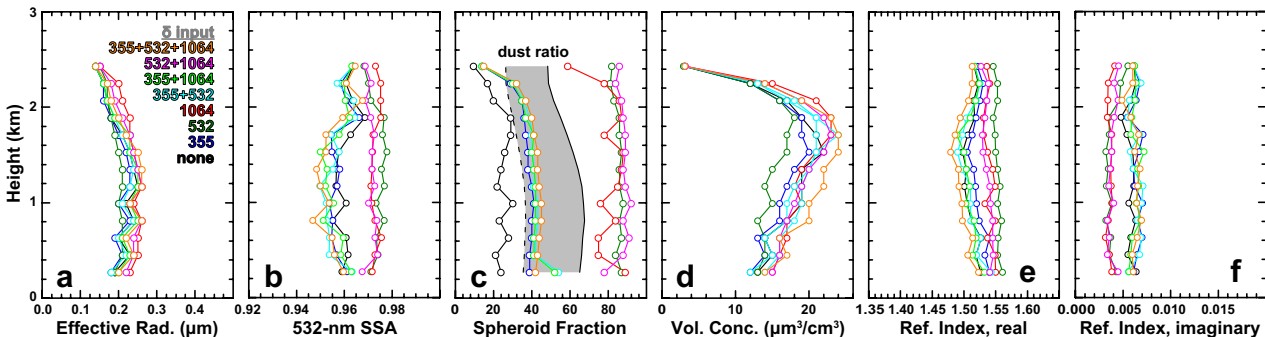

**Figure 5.** Same as Figure 3 but for the input data presented in Figure 4. The dashed line refers to the profile of the dust mixing ratio obtained according to *Burton et al.* (2012).

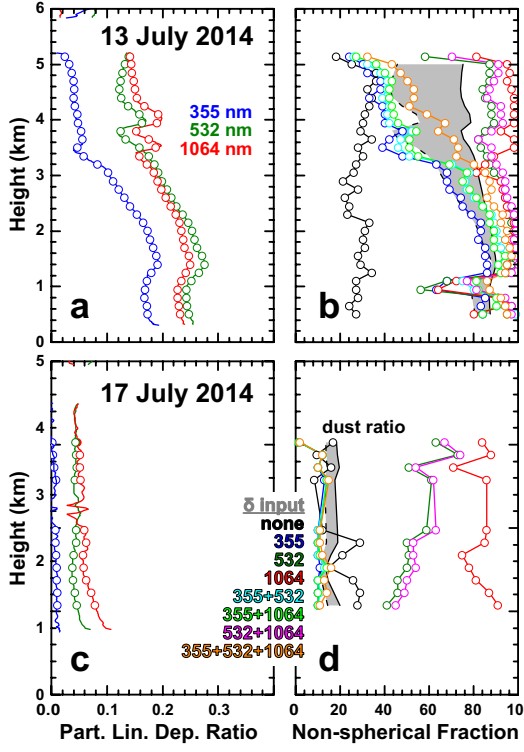

**Figure 6.** Profiles of (a and c) $\delta_\lambda$ and (b and d) the retrieved non-spherical fraction for DISCOVER-AQ Colorado flights on 13 and 17 July 2014. Note that values differ from Figure 5 in *Burton et al.* (2015) as we have used a longer averaging period of 23 minutes in our work (see Table 1).

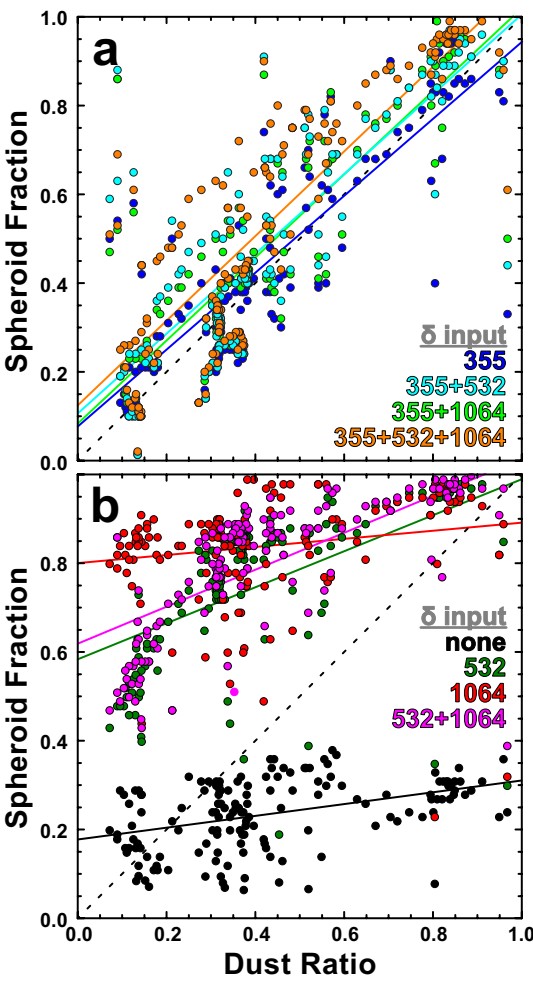

**Figure 7.** Connection between the retrieved spheroid fraction (from inversion) and the ratio of non-spherical particles to the 532-nm backscatter coefficient (from lidar measurements of $\delta_{532}$) for the input data sets listed in Table 2 and the cases listed in Table 1.

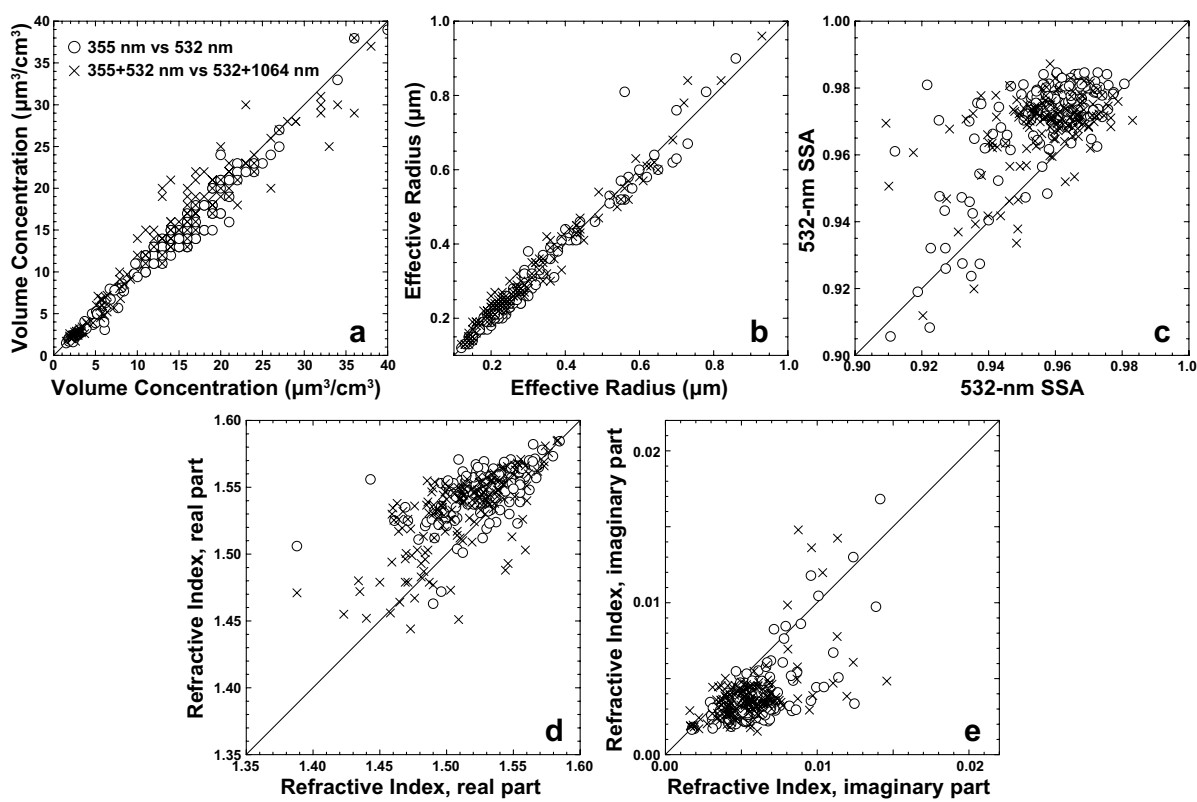

**Figure 8.** Correlation of (a) volume concentration, (b) effective radius, (c) 532-nm SSA, and (d) real and (e) imaginary part of the refractive index obtained from inversion runs that use a single particle linear depolarization ratio at 355 nm or 532 nm (3+2+1, circles) or two particle linear depolarization ratios at 355 and 532 nm or 532 and 1064 nm (3+2+2, crosses) as input data.

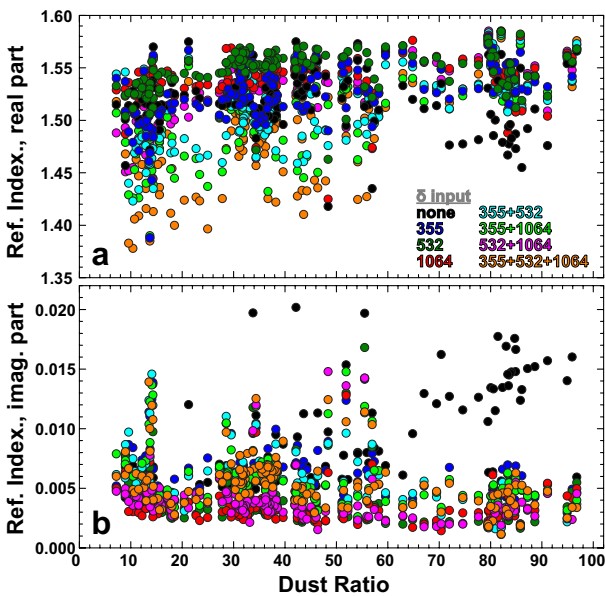

**Figure 9.** The retrieved (a) real and (b) imaginary parts of the refractive index and their connection to the dust ratio for the different input data sets.

**Table 1.** Overview of the 3+2+3 lidar measurements taken with BERTHA and HSRL-2 and used in this study. The HSRL-2 aerosol type was determined following the procedure outlined in *Burton et al.* (2012). Note that HSRL-2 measurements include transit flights. The lower and upper values for the range of dust ratios refer to values obtained from using methods described by *Tesche et al.* (2009b) and *Mamouri and Ansmann* (2014) (sum of fine and coarse dust), respectively.

| Date | Time (UTC) | Height (km) | mean $\delta_{532}$ | Dust ratio (%) | aerosol type |
|---|---|---|---|---|---|
| **HSRL-2: DISCOVER-AQ California (2013), Texas (2013), Colorado (2014)** | | | | | |
| 20130130 | 1656 - 1712 | 0.3 - 1.0 | 0.05±0.04 | 16–65 | urban/pollution, fresh smoke |
| | | 1.0 - 1.2 | 0.28±0.04 | 89–90 | dusty mix |
| 20130208 | 1737 - 1802 | 2.0 - 2.4 | 0.32±0.04 | 100–94 | dusty mix |
| | | 3.8 - 4.2 | 0.12±0.01 | 40–70 | dusty mix, urban/pollution |
| 20130925 | 2057 - 2105 | 0.3 - 2.4 | 0.10±0.01 | 35–61 | dusty mix, urban/pollution |
| 20130926 | 2036 - 2041 | 0.3 - 2.1 | 0.10±0.01 | 35–60 | dusty mix, urban/pollution |
| 20130928 | 1612 - 1617 | 0.3 - 1.9 | 0.04±0.01 | 14–19 | urban/pollution |
| 20140713 | 1435 - 1446 | 0.4 - 3.0 | 0.10±0.03 | 34–56 | dusty mix, urban/pollution |
| | 1713 - 1736 | 0.5 - 5.1 | 0.20±0.05 | 67–83 | dusty mix |
| 20140717 | 1917 - 1919 | 2.0 - 4.0 | 0.04±0.00 | 13–18 | urban/pollution, polluted marine |
| 20140722 | 2009 - 2036 | 2.0 - 3.0 | 0.15±0.02 | 52–76 | dusty mix |
| | | 3.0 - 5.5 | 0.10±0.01 | 31–53 | urban/pollution, dusty mix |
| **BERTHA: SALTRACE, Barbados** | | | | | |
| 20140303 | 2230 - 2330 | 1.0 - 2.8 | 0.12±0.04 | 40–61 | dusty mix |
| 20140620 | 2310 - 0210 | 1.0 - 4.0 | 0.26±0.02 | 83–88 | mineral dust |

**Table 2.** Combinations of $\delta_\lambda$ that were used as inversion input in addition to the conventional 3+2 data set (Set I) together with the mean number of averaged solutions, statistics on the considered discrepancies, and median absolute errors (from averaging of the considered solutions) of the retrieved effective radius ($r_{eff}$), single-scattering albedo (SSA), spheroid fraction (SF), volume concentration ($v$), and real and imaginary part of the refractive index ($m_r$ and $m_i$) for the respective input data sets. The bottom part of the table provides the slope, intercept, and squared correlation coefficients ($R^2$) for the linear fits between dust ratio (as obtained with the method of *Tesche et al.* 2009b) and spheroid fraction presented in Figure 7.

| Data set | I | II | III | IV | V | VI | VII | VIII |
|---|---|---|---|---|---|---|---|---|
| 355 nm | - | X | - | - | X | X | - | X |
| 532 nm | - | - | X | - | X | - | X | X |
| 1064 nm | - | - | - | X | - | X | X | X |
| solutions | 163 | 157 | 158 | 159 | 160 | 159 | 160 | 164 |
| **discrepancy** | | | | | | | | |
| mean | 3.77 | 7.85 | 8.97 | 11.74 | 14.72 | 16.03 | 13.39 | 20.58 |
| median | 1 | 7 | 8 | 11 | 15 | 17 | 13 | 22 |
| range | 0–23 | 2–30 | 1–30 | 1–35 | 3–35 | 3–36 | 3–37 | 4–41 |
| **absolute error, median value** | | | | | | | | |
| $r_{eff}$ | 0.043 | 0.045 | 0.036 | 0.052 | 0.054 | 0.066 | 0.061 | 0.071 |
| SSA | 0.036 | 0.034 | 0.023 | 0.027 | 0.034 | 0.035 | 0.029 | 0.038 |
| SF | 24 | 5 | 6 | 11 | 5 | 6 | 6 | 6 |
| $v$ | 4 | 4 | 3 | 5 | 4 | 6 | 4 | 6 |
| $m_r$ | 0.068 | 0.059 | 0.046 | 0.059 | 0.065 | 0.070 | 0.062 | 0.071 |
| $m_i$ | 0.006 | 0.005 | 0.003 | 0.004 | 0.005 | 0.005 | 0.004 | 0.005 |
| **linear fits, Figure 7** | | | | | | | | |
| Slope | 0.13 | 0.87 | 0.41 | 0.09 | 0.94 | 0.90 | 0.42 | 0.95 |
| Intercept | 18 | 8 | 58 | 80 | 9 | 11 | 62 | 13 |
| $R^2$ | 0.17 | 0.69 | 0.33 | 0.03 | 0.73 | 0.68 | 0.46 | 0.73 |