# Peer review of "3+2+X: What is the most useful depolarisation input for retrieving microphysical properties of non-spherical particles from lidar measurements using the spheroid model of Dubovik et al. (2006)?"

_Atmospheric Measurement Techniques, 2019_

## Referee Comment (RC1) · Anonymous Referee #2 · 22 Apr 2019

Authors present results, related to inversion of lidar observations, containing three depolarization ratios to the particle microphysics. This is interesting and important study. Paper is clearly written, contains new results and can be recommended for publication. Still I think that the manuscript will become better if authors address several important points. 1. Authors use the spheroids model to mimic the scattering properties of dust or other non-spherical particles. Thus, the analysis presented is valid only for this model. By my opinion, the manuscript should start with consideration, how well the spheroid

model describes depolarizing properties of real aerosols. I would suggest to compare available in literature depolarization measurements with model predictions. Some information can be found in: Miffre et al., Journal of Quantitative Spectroscopy & Radiative Transfer 169 (2016) 79–90 Mamouri et al., Atmos. Meas. Tech., 10, 3403–3427, 2017 (this one is cited by authors) The issues related to modeling of the smoke depolarization ratio are discussed also in Mishchenko et al., Appl Opt 55, No. 35, 2016. 2. For convenience of results discussion, I would suggest to add in the beginning a kind of a table (or figure), showing depolarization ratios, predicted by spheroid model for realistic PSD of dust for pure spheroids and for spheroids fraction 40%, 60%....It will help in the analysis of results. It is important to show depolarizations for different values of the imaginary part (mI). From my knowledge even pure spheroids have difficulty in reproducing high depolarization ratios, especially for high mI. 3. Authors assume spectrally independent the particle refractive index. In reality, in dust the imaginary part of the refractive index increases in UV, leading to fast decrease of depolarization ratio. This should be discussed.

Technical comments. p.5 ln 27 "This case represents nearly pure dust conditions..." It is worth to mention that this is long-transported dust, which can be modified during transportation. p.6.ln.6 The inversion of all input data sets shows a decrease of effective radius with height" Why? If Angstrom and depolarization are stable... p.6. ln 12 "In fact, non-spherical particle fractions were never found to exceed 40% when using the traditional 3+2 input" Inverse problem for 3+2 set is strongly underdetermined, so retrieved spheroid fraction is un-trustable. This is why usually 100% of spheroids is assumed. p.6 ln9 "The high values of _532 lead to a dust fraction above 80% " Here the table with predicted depolarizations would be useful. What discrepancy did you obtain in retrievals? Fig.2. Why not to show retrieved real and imaginary parts? Error bars should be added.

p.11 ln.18 " The most realistic non-spherical fractions is found when using depolarization information at 355 nm." Have you any explanation for this?

---

## Referee Comment (RC2) · Anonymous Referee #1 · 1 May 2019

The paper is appropriate for AMT and a good contribution to the literature of lidar methods and technologies.

However, improvements are needed. The discussion needs to be extended.

Throughout the paper it is assumed that the fine mode (particles with diameter < 1 micrometer) contains ONLY spherical (anthropogenic) particles. But in reality, the fine mode also contains non-spherical dust particles and may contain also non-spherical

smoke particles (pronounced accumulation mode). This can be concluded from the Burton et al. (2015) paper with a strong wavelength dependence of the depolarization ratio found for dry smoke particles in the upper troposphere. It was confirmed by Hu et al, (2019) and Haarig et al. (2018) for the 2017 stratospheric smoke event. All this is not considered in the paper. Thus, the recommendations given in the paper regarding the configuration of a multiwavelength polarization lidar are of limited use. It must be clearly stated for what scenarios the conclusions hold! Aerosol scenarios with a large non-spherical fine mode fraction are not considered!

And the other important point is that the study here is based on a spheroidal dust model. As long as a trustworthy dust shape model is not available, all the simulations, all the inversion results, are just speculation! This must be repeated more frequently throughout the text. There is simply no solid conclusion what configuration the best is, except 3+2+3!

Some details:

P2, L28-29: All these inversion methods with spheroidal dust particles are not convincing. Yes, the Tesche 2009, 2011 way is much more convincing! Why is the alternative concept (dust/nondust separation by means of the particle depolarization ratio) approach not discussed in more detail? There is also the next-step approach by Mamouri et al (2016 and 2017) to circumvent the spheroidal shape problems. A more complete discussion is needed. What does it help to have even a 3+3+3 system when we need to base the full concept on the questionalbe dust spheroidal model?

P4, L2: In the Hu et al 2019 paper (published in 2019), the extinction profiles are computed from the elastic channels only. No Raman lidar solutions, no 3+2+3. On the other hand, here you kindly provide these references on the dry smoke observations, Hu, Haarig! But then you ignore all these realistic aerosol scenarios in the rest of the manuscript.

P5, L16: Table 1 corroborates my opinion! The considered non-dust part is just spherical fine mode aerosol (urban haze, smoke). The paper does not cover the full reality of aerosol scenarios.

P9, L2-6: We need such statements more frequently!

P9, L7-L17: All this is confusing! Again, the Shin et al (2018) results are based on the spheroidal model. So why should the results agree with observations? And lidars do not measure lidar ratios and depol ratios at 870 and 1020 nm! Gasteiger played around with many shape configurations and failed because there is no realistic shape model for dust available. At least Gasteiger showed how sensitive all the modelled dust optical properties depend on particle shape.

P9, L18-27: The authors performed ... systematic investigations ... with a wrong shape model! ....in contradiction with Gasteiger and Freudenthaler! Sure, what did you expect?

At the end, the discussion gives the impression: Because we do not have a real alternative, we take the spheroidal model, and assume that all this is quite ok! But it is not ok!

Figure 2, and also P11, L15: Tesche et al 2009 is not the end of the street (separation of dust and non dust with one depol ratio). The Mamouri 2014 and 2017 papers extended the approach towards fine dust and coarse dust separation... That should be mentioned! They used different depol values for fine mode and coarse mode. And this is obviously the reality! And then you will get higher dust fractions than the one shown in Figure 2 by using the Tesche 2009 approach. All this is not mentioned in the manuscript. So the discussion is incomplete. Fortunately, it is considered in the inversion models (dependence of the depolarization ratio on particle size). The size dependence was already shown by Gasteiger and in many other papers and are now even measured with polarization OPCs (Tian et al., ACP 2018). However, one has to be careful, POPC measure at 120 deg, lidar at 180 deg. But the size dependence of the depol is similar to what is measured for example by Sakai, Appl. Opt 2010 or in the

AIDA KIT chamber... at 180deg.

Concerning the figures: It would be good to have the legend I: -, II: 355 nm, III: 532 nm always shown in all figures (as a tall table with 8 lines below each other). It will not take much space. I had to take a piece of paper, wrote down all the scenarios to have this information always present when stepping through all the figures....

Confusing Figure 6! The message is obvious, but the scatter in the results is large as well.

Confusing Figure 7: Set II / Set V can easily be interpreted as ratio.... One has to read all the text in the caption and in the figure carefully before looking at the results. And then again, the questions: what was Set I and what was Set V, and so on....?

Then Figure 8, .... even a further decrease in information, no x-axis and y-axis text anymore, just the hint: same as in Figure 7.

So, all in all, it was really not easy to review this paper and these figures... and thus: It was not a pleasure! So, please improve all this significantly.

---

## Referee Comment (RC3) · Anonymous Referee #3 · 8 May 2019

This work is valuable since it discusses the most useful depolarization input for retrieving dust properties from lidar measurements using the spheroid model of Dubovik et al. (2006). The Dubovik model was generated for retrieving dust microphysical properties with passive remote sensing, but it has been extensively used for dust retrievals with lidars. A discussion relevant on the Dubovik model capabilities and limitations for dust retrievals using lidar measurements is missing from the literature. Although the work in discussion does not extensively do this, it opens a discussion for the most useful input

in terms of depolarization at 355, 532 and 1064 nm.

Unfortunately the paper is very confusing since the authors present their results as if they consider a universal spheroid model. The spheroid model of Dubovik et al. (2006) cannot be considered as a universal spheroid model, due to the assumption used for the aspect ratio distribution and the mixture of prolate and oblate spheroids. This is discussed in Dubovik et al. (2006), showing that the values of P22/P11 (i.e. the phase matrix elements relevant to depolarization) can be different for different aspect ratio distributions (see Fig. 12 in Dubovik et al. (2006); the values at 180 degrees are not shown but from the scattering angles up to ∼170 degrees the tendency is clear). Dubovik et al. state in page 15 of their paper that: "Figure 12 illustrates the phase matrices simulated for desert dust aerosol using an ensemble of spheroids differing only by their axis ratio distributions. .... Some differences can be seen for ... P22/P11, however they are likely to be insignificant for passive remote sensing applications."

This is most probably the reason the results of the work in discussion are different from results shown in Gasteiger and Freudenthaler (2014), who assumed the spheroidal shape for non-spherical particles, but assumed different aspect ratios and mixture of prolate and oblate particles, than the Dubovik model.

In conclusion, although the work in discussion can be a valuable tool for dust lidar retrievals, it should undergo major revisions before it is published in AMT. Specifically, the authors should shift its objective to discuss what their results are actually about, starting with changing the title to something similar to: "What are the most useful depolarization input for retrieving the microphysical properties of dust particles from lidar measurements using the spheroid model of Dubovik et al. (2006)". Please revise the whole manuscript accordingly.

Last, please note that the focus of the work is on dust particles and not on "non-spherical particles" in general. Moreover (a) change the retrieved "non-spherical fraction" to "spheroid fraction" throughout the document, (b) include plots of the retrieved

real and imaginary part of the refractive index in the results section and (c) provide a more extensive discussion on the effect of the assumption of spectrally-independent refractive index used in the analysis.

---

## Author Comment (AC1) · 25 Jun 2019

Interactive comments on "3 + 2 + X: What is the most useful depolarization input for retrieving microphysical properties of non-spherical particles from lidar measurements by assuming spheroidal particle shapes?" by Matthias Tesche et al.

Referee comments are noted in black. Our replies are given in blue.

We would like to thank all Referees for their constructive comments. Please find our point-by-point replies below. A list of added references is given at the end of this reply letter. We have also attached a revised version of the manuscript with all changes marked.

**Anonymous Referee #1**

The paper is appropriate for AMT and a good contribution to the literature of lidar methods and technologies.

However, improvements are needed.

The discussion needs to be extended. Throughout the paper it is assumed that the fine mode (particles with diameter < 1 micrometer) contains ONLY spherical (anthropogenic) particles. But in reality, the fine mode also contains non-spherical dust particles and may contain also non-spherical smoke particles (pronounced accumulation mode). This can be concluded from the Burton et al. (2015) paper with a strong wavelength dependence of the depolarization ratio found for dry smoke particles in the upper troposphere. It was confirmed by Hu et al, (2019) and Haarig et al. (2018) for the 2017 stratospheric smoke event. All this is not considered in the paper. Thus, the recommendations given in the paper regarding the configuration of a multiwavelength polarization lidar are of limited use. It must be clearly stated for what scenarios the conclusions hold! Aerosol scenarios with a large non-spherical fine mode fraction are not considered!

The assume that the Referee is referring to the dust ratio as inferred from the measurements of the particle linear depolarisation ratio and used to assess the retrieved spheroid fractions. In that case, the Referee is correct that our study assumes that coarse-mode dust is mixed with a spherical fine-mode aerosol. This is now specifically addressed in the new Section 3.2 *Retrieval of dust fraction from optical data*. As to the inversion routine, we would like to clarify that the spheroid fraction as first used by Veselovskii et al. (2010) does not depend on radius and accounts for both the fine and coarse mode. Therefore, cases with non-spherical particles in the fine mode can be properly described by mathematical module we use in the inversion routine Any solution described by a high value of spheroid fraction and a pronounced fine mode (in particular, monomodal particle size distributions) refers to exactly the aerosol scenarios mentioned by the Referee.

And the other important point is that the study here is based on a spheroidal dust model. As long as a trustworthy dust shape model is not available, all the simulations, all the inversion results, are just speculation! This must be repeated more frequently throughout the text. There is simply no solid conclusion what configuration the best is, except 3+2+3!

We have added revisions throughout the text to better emphasise two points, as also suggested by the other Referees: (i) that the Dubovik model is not a universal spheroid model as it

applies a fixed axis-ratio distribution and a specific oblate-to-prolate ratio for the spheroids and (ii) that our conclusions only hold for the application of this particular model.

Some details:

P2, L28-29: All these inversion methods with spheroidal dust particles are not convincing. Yes, the Tesche 2009, 2011 way is much more convincing! Why is the alternative concept (dust/non-dust separation by means of the particle depolarization ratio) approach not discussed in more detail? There is also the next-step approach by Mamouri et al (2016 and 2017) to circumvent the spheroidal shape problems. A more complete discussion is needed. What does it help to have even a 3+3+3 system when we need to base the full concept on the questionable dust spheroidal model?

The Referee is rightfully criticising that the text on the aerosol-type separation methodologies has not been sufficient. We have expanded the description of the alternative concept by introducing a new Section 3.2 Retrieval of dust fraction from optical data. Finally, we have changed the dust ratio values in Table 1 to a range of likely dust contribution as obtained from following *Tesche et al.* (2009b) and *Mamouri and Ansmann* (2014; 2017) for the lower and upper limit, respectively.

The text in the introduction has been revised to: "*Scattering kernels based on Mie theory cannot represent light scattering by non-spherical particles, i.e. particles that lead to increased $\delta_\lambda$ in a lidar measurement. A way to circumvent this problem is to split the optical input according to the information related to spherical and non-spherical particles. The thus obtained data sets can subsequently be used to run the inversion considering only spherical scatterers (i.e. Mie kernels) and non-spherical scatterers (i.e. spheroid kernels), respectively. We provide a detailed discussion of this aspect in Section 3.3. However, the aim of this work is to gain insight into the performance of the inversion using Dubovik's model for mixed dust cases as such scenarios have not yet been considered in earlier studies.*"

The new text in Section 3.2 Retrieval of dust fraction from optical data is: "*The particle linear depolarisation ratio is an intensive aerosol property that can be applied for aerosol classification* (Burton et al., 2012; Groß et al., 2013. *Because of its sensitivity to particle shape, it can also be used to separate the contributions of spherical and non-spherical particles to the optical parameters measured with aerosol lidar* (Tesche et al., 2009b; Burton et al., 2014) *or sun photometer* (Shin et al., 2019). *This approach study generally assumes mixtures with a coarse-mode that is composed of mineral dust and a spherical fine-mode.* Tesche et al. (2009b, 2011b) *use measurements of $\delta_{532}$ together with threshold values representative for pure aerosol types to separate the contribution of dust and biomass-burning smoke to the optical properties measured with multiwavelength aerosol Raman lidar at Cape Verde. Their approach assumes an external mixture of two aerosol types. A generalised form of this method that covers a broader variety of aerosol mixtures has been presented by* Burton et al. (2014) *for measurements with HSRL-1.* Mamouri and Ansmann (2014; 2017) *have refined the aerosol-type separation further using a two-step approach that allows for the separation of contributions of coarse dust, fine dust, and a non-dust aerosol type, i.e. marine or continental aerosol.*

*In principle, these aerosol-type separation techniques can be used to obtain input data sets for the inversion of lidar data that represent the spherical and non-spherical particles in a mixed aerosol plume, respectively. The inversion could then be run with the conventional 3+2*

*input data set and the spheroid fraction (see Section 3.3) set to either 0% (i.e. Mie kernels) or 100%. In this study, however, we aim to test how the inversion performs with different combinations of additional depolarisation-ratio input to account for the contribution of non-spherical particles to the optical input data. We use the dust ratio, i.e. the ratio of dust-related to total backscatter coefficient at 532 nm, as an estimate of the dust contribution and for comparison to the spheroid fraction inferred from the inversion. Dust ratios were either taken from the DISCOVER-AQ data base (these values have been derived according to* Burton et al. (2014)*), or calculated following* Tesche et al. (2009b) *(with $\delta_d$ = 0.32 and $\delta_{nd}$ = 0.01 to replicate the values from HSRL-2) and* Mamouri and Ansmann (2014; 2017) *($\delta_{dc}$ = 0.39, $\delta_{df}$ = 0.16, and $\delta_{nd}$ = 0.02 to obtain the contributions of fine and coarse dust). In the discussion of our findings, we will consider the dust ratio for the two-component* (Tesche et al.,2009b; Burton et al., 2014) *and tree-component* (Mamouri and Ansmann, 2014; 2017) *mixtures as lower and upper limit, respectively, of the likely dust contribution."*

P4, L2: In the Hu et al 2019 paper (published in 2019), the extinction profiles are computed from the elastic channels only. No Raman lidar solutions, no 3+2+3.

We are sorry for the mistake. The focus of this statement was supposed to be on the capability for triple-wavelength measurements of the particle linear depolarisation ratio. The statement has been revised for clarity : *"To date, few lidar instruments have the capability to measure particle linear depolarisation ratios at three wavelengths simultaneously and we refer to Burton et al. (2015), Haarig et al. (2017a), and Hu et al. (2019)."*

On the other hand, here you kindly provide these references on the dry smoke observations, Hu, Haarig! But then you ignore all these realistic aerosol scenarios in the rest of the manuscript.

Following the suggestions of all Referees, we have introduced a new Section 2: *Spectral δ for mineral dust from measurements and modelling* and a new Figure 1 to provide a better overview over field and lab measurements of spectral particle linear depolarisation ratios as well as findings from modelling using spheroids and irregularly shaped particles. The issue that large particle linear depolarisation ratios have also been observed for biomass-burning smoke is now being discussed in this section. Nevertheless, such observations are still scarce. In addition, the focus of our study is on mixtures that include mineral dust. We are therefore confident that highly depolarising dried smoke particles don't have a large impact on our investigation.

New Figure 1 looks like this:

[revised manuscript text omitted]

P5, L16: Table 1 corroborates my opinion! The considered non-dust part is just spherical fine mode aerosol (urban haze, smoke). The paper does not cover the full reality of aerosol scenarios.

The values of the dust ratio given in Table 1 have been revised to now cover a range as described in new Section 3.2 (see also the detailed reply to the Referee's second comment).

P9, L2-6: We need such statements more frequently!

Revisions have been made throughout the manuscript to emphasise that we are considering the specific spheroidal model of *Dubovik et al.* (2006) which has not been developed primarily for lidar applications.

P9, L7-L17: All this is confusing! Again, the Shin et al (2018) results are based on the spheroidal model. So why should the results agree with observations? And lidars do not measure lidar ratios and depol ratios at 870 and 1020 nm! Gasteiger played around with many shape configurations and failed because there is no realistic shape model for dust available. At least Gasteiger showed how sensitive all the modelled dust optical properties depend on particle shape.

*We have added the new Section 2: Spectral $\delta$ for mineral dust from measurements and modelling and a new Figure 1 to provide a better overview over field and lab measurements of spectral particle linear depolarisation ratios as well as findings from modelling using spheroids and irregularly shaped particles. We hope that this Section can provide the context for discussing the depolarisation ratios inferred from AERONET measurements (Shin et al., 2018) and a more universal spheroid model (Gasteiger and Freudenthaler, 2014). We have also revised the particular paragraph to emphasise the differences connected to the different findings. We believe that it is important to present the available data even though one might not expect them to be compatible in the first place. The revised text in the discussion section now reads: "The results we obtain from our study are somewhat contradictory to the findings of* Gasteiger and Freudenthaler (2014) *and* Shin et al. (2018) *who attribute the greatest informational value and representativeness to observations of $\delta_{1064}$. This is likely due to the difference in the setup of the different studies. On the one hand,* Gasteiger and Freudenthaler (2014) *applied a spheroid model that was constrained in neither the aspect ratio not the oblate-to-prolate ratio of the spheroids. On the other hand,* Shin et al. (2018) *are referring to AERONET data which have been derived for scattering angles smaller than 180° (and extrapolated to the backscatter direction) and represent values for the entire atmospheric column. However, the strong and weak effects of using $\delta_{355}$ and $\delta_{1064}$ , respectively, in the inversion of lidar measurements of dust-containing aerosol layers based on the spheroid model of* Dubovik et al. (2006) *also indicate that the model's constraints on the aspect-ratio distribution and the ratio of oblates to prolates have a strong effect on making full use of the informational content provided at different wavelenths in lidar applications. We stress again that the conclusions of this study are valid only for the inversion of lidar data that resorts to describe the light-scattering properties of non-spherical dust particles by means of the spheroid model of* Dubovik et al. (2006) *and that any finding might be strongly related to the weaknesses of this particular light-scattering model* (Müller et al., 2010, 2013). *Nevertheless, no other model has been applied as widely in the inversion of lidar data* (Veselovskii et al., 2010; Di Girolamo et al., 2012; Papayannis et al., 2012; Müller et al., 2013). *In addition, there has so far been no systematic investigation of the usefulness of different depolarisation input for this particular inversion setup. We therefore believe that this work will contribute to a better understanding of the usefulness and limitations of the model of* Dubovik et al. (2006) *in lidar applications as well as to further emphasise the need for a more general model to describe light-scattering by non-spherical particles at very large scattering angles.*

*Following on the initial work of* Veselovskii et al. (2010)*, we have performed the first systematic investigation of the effect of all possible combinations of depolarization-related inversion input at the wavelengths of 355, 532, and 1064nm on the retrieved aerosol microphysical properties. The aim of this study is to assess the performance of the inversion procedure to lidar measurements conducted in the presence of mixture of spherical and non-spherical particles. So far, inversions on the presence of mineral dust have only been attempted if pure-dust conditions could be assumed* (Veselovskii et al., 2010; Di Girolamo et al., 2012) *or of the contribution of the non-spherical scatterers had been screened from the optical input data* (Tesche et al., 2011a, b).*"*

P9, L18-27: The authors performed...systematic investigations...with a wrong shape model!....in contradiction with Gasteiger and Freudenthaler! Sure, what did you expect?

As stated in the reply to the previous comment, we have revised the text for stronger emphasis on the differences between the different approached.

At the end, the discussion gives the impression: Because we do not have a real alternative, we take the spheroidal model, and assume that all this is quite ok! But it is not ok!

The Referee is certainly correct that the spheroidal Dubovik model does not mark the end of the road when it comes to lidar applications. But despite its known limitations and certainly also because of a lack of alternatives, it is being applied for the inversion of lidar data as we have mentioned in the introduction to the paper. We therefore think that it is important to present an investigation of the effect of using all possibilities of different input data for the inversion using the Dubovik model. Such a relational investigations has not yet been presented. We do not state that all is quite okay. In the revised manuscript we state repeatedly that the presented findings are only applicable to the specific inversion setup we have used in our study. Following the suggestion of another Referee we have also revised the title of our manuscript to clarify that the reader should not expect a universal solution to the problem of light-scattering by non-spherical particles from this work. We have also added the following text in the third paragraph of the discussion to provide further context for our work: "*However, the strong and weak effects of using $\delta_{355}$ and $\delta_{1064}$ , respectively, in the inversion of lidar measurements of dust-containing aerosol layers based on the spheroid model of* Dubovik et al. (2006) *also indicate that the model's constraints on the aspect-ratio distribution and the ratio of oblates to prolates have a strong effect on making full use of the informational content provided at different wavelenths in lidar applications. We stress again that the conclusions of this study are valid only for the inversion of lidar data that resorts to describe the light-scattering properties of non-spherical dust particles by means of the spheroid model of* Dubovik et al. (2006) *and that any finding might be strongly related to the weaknesses of this particular light-scattering model (Müller et al., 2010, 2013). Nevertheless, no other model has been applied as widely in the inversion of lidar data* (Veselovskii et al., 2010; Di Girolamo et al., 2012; Papayannis et al., 2012; Müller et al., 2013). *In addition, there has so far been no systematic investigation of the usefulness of different depolarisation input for this particular inversion setup. We therefore believe that this work will contribute to a better understanding of the usefulness and limitations of the model of* Dubovik et al. (2006) *in lidar applications as well as to further emphasise the need for a more general model to describe light-scattering by non-spherical particles at very large scattering angles.*"

Figure 2, and also P11, L15: Tesche et al 2009 is not the end of the street (separation of dust and non-dust with one depol ratio). The Mamouri 2014 and 2017 papers extended the approach towards fine dust and coarse dust separation...That should be mentioned! They used different depol values for fine mode and coarse mode. And this is obviously the reality! And

then you will get higher dust fractions than the one shown in Figure 2 by using the Tesche 2009 approach. All this is not mentioned in the manuscript. So the discussion is incomplete. Fortunately, it is considered in the inversion models (dependence of the depolarization ratio on particle size). The size dependence was already shown by Gasteiger and in many other papers and are now even measured with polarization OPCs (Tian et al., ACP 2018). However, one has to be careful, POPC measure at 120 deg, lidar at 180 deg. But the size dependence of the depol is similar to what is measured for example by Sakai, Appl. Opt 2010 or in the AIDA KIT chamber...at 180deg.

This issue of the aerosol-type separation is now addressed in more detail in new Section 3.2 (see also the detailed reply to the Referee's second comment). In addition, we have revised the figures (now Figures 3, 5, and 6) to show a range of likely dust ratios with *Tesche et al.* (2009b) as lower and fine plus coarse dust following *Mamouri and Ansmann* (2014; 2017) as upper limit rather than a line.

Concerning the figures: It would be good to have the legend I: -, II: 355 nm, III: 532 nm always shown in all figures (as a tall table with 8 lines below each other). It will not take much space. I had to take a piece of paper, wrote down all the scenarios to have this information always present when stepping through all the figures....

The legends of all figures have been changed from Set name to the values of the used wavelengths.

Confusing Figure 6! The message is obvious, but the scatter in the results is large as well.

We don't know what to reply to this comment. We are aware that the figures are somewhat hard to understand instantly without reading the explanations in the text. We have revised the figures and figure captions to ease the accessibility of the figure content.

Confusing Figure 7: Set II / Set V can easily be interpreted as ratio.... One has to read all the text in the caption and in the figure carefully before looking at the results. And then again, the question: what was Set I and what was Set V, and so on....?

The figure (now Figure 8) has been revised to clearly state the wavelengths and to avoid the interpretation that ratios are shown. In addition, we have added results for the complex refractive index as suggested by the other Referees.

Then Figure 8,.... even a further decrease in information, no x-axis and y-axis text anymore, just the hint: same as in Figure 7.

The figure has been removed from the manuscript.

So, all in all, it was really not easy to review this paper and these figures...and thus: It was not a pleasure! So, please improve all this significantly

We are very sorry to hear this. We hope that our revisions have improved the reading experience to the Referee's satisfaction.

**Anonymous Referee #2**

Authors present results, related to inversion of lidar observations, containing three depolarization ratios to the particle microphysics. This is interesting and important study. Paper is clearly written, contains new results and can be recommended for publication.

Still I think that the manuscript will become better if authors address several important points.

1. Authors use the spheroids model to mimic the scattering properties of dust or other non-spherical particles. Thus, the analysis presented is valid only for this model. By my opinion, the manuscript should start with consideration, how well the spheroid model describes depolarizing properties of real aerosols. I would suggest to compare available in literature depolarization measurements with model predictions. Some information can be found in:
- Miffre et al., Journal of Quantitative Spectroscopy & Radiative Transfer 169 (2016) 79–90
- Mamouri et al., Atmos. Meas. Tech., 10, 3403–3427, 2017 (this one is cited by authors)
- The issues related to modeling of the smoke depolarization ratio are discussed also in Mishchenko et al., Appl Opt 55, No. 35, 2016.

*Following the suggestions of all Referees, we have introduced a new Section 2: Spectral δ for mineral dust from measurements and modelling and a new Figure 1 to provide a better overview over field and lab measurements of spectral particle linear depolarisation ratios as well as findings from modelling using spheroids and irregularly shaped particles. The issue that large particle linear depolarisation ratios have also been observed for biomass-burning smoke is now being discussed in this section.*

*New Figure 1 looks like this:*

[revised manuscript text omitted]

2. For convenience of results discussion, I would suggest to add in the beginning a kind of a table (or figure), showing depolarization ratios, predicted by spheroid model for realistic PSD of dust for pure spheroids and for spheroids fraction 40%, 60%....It will help in the analysis of results.

The presentation of particle linear depolarisation ratios for realistic size distributions and different spheroid fractions is a very good suggestion. However, we believe that the effort involved to get meaningful results from such an approach would warrant a separate study entirely as it goes well beyond the scope of this paper. Instead, we have added the discussion of field and lab measurements of the particle linear depolarisation ratio as well as of modelling studies with focus on lidar applications in new Section 2 (see reply to previous comment). In any case, a sensitivity as suggested by the Referee has been presented by *Veselovskii et al.* (2010), i.e. in their Figures 8 and 14.

It is important to show depolarizations for different values of the imaginary part (mI). From my knowledge even pure spheroids have difficulty in reproducing high depolarization ratios, especially for high mI.

We have now added a detailed presentation of the different inversion results for the refractive index to Figures 3, 5, 8, 9, and a new Figure 10. The issue addressed by the Referee has also been shown before by *Veselovskii et al.* (2010).

3. Authors assume spectrally independent the particle refractive index. In reality, in dust the imaginary part of the refractive index increases in UV, leading to fast decrease of depolarization ratio. This should be discussed.

The Referee is correct. We have added the following statement to Section 3.3 to clarify the setup of the inversion algorithm with respect to the refractive index: "*The inversion uses a single refractive index that is independent of particle size and wavelength.*"

Regarding the effect of this constraint, we have added the following paragraph top the discussion of the findings: "*The inversion assumes a spectrally independent complex refractive index. In contrast, mineral dust is known to show a strong increase in the imaginary part of the refractive index with smaller wavelengths. This issue has been explored by* Veselovskii et al. (2010) *who conclude that (i) the error of the volume concentration is estimated as 17% to 25% depending on the contribution of large particles and (ii) a fixed imaginary part refers to the mean value of the spectrally dependent imaginary part. A detailed investigation of the assumption of spectrally independent refractive indices is beyond the scope of this study.*"

We are afraid that an in-depth investigation of the effect of using a spectrally independent refractive index in the inversion of lidar data goes well beyond the scope of this study. In fact, this issue warrants its own dedicated study.

**Technical comments.**

p5, ln27: "This case represents nearly pure dust conditions..." It is worth to mention that this is long-transported dust, which can be modified during transportation.

The following statement has been added to the text: "*However, the dust arrived at Barbados after several days of long-range transport. Its bulk properties might have been modified during transport compared to freshly emitted dust.*"

p6, ln6: The inversion of all input data sets shows a decrease of effective radius with height" Why? If Angstrom and depolarization are stable...

We are sorry for showing only parts of the optical data set. We have now added plots of the lidar ratio and the Angstrom exponent to Figure 2 (former Figure 1). These show an increase of the lidar ratio and the extinction-related Angstrom exponent that is in line with a decrease of the effective radius.

p6, ln12: "In fact, non-spherical particle fractions were never found to exceed 40% when using the traditional 3+2 input" Inverse problem for 3+2 set is strongly underdetermined, so retrieved spheroid fraction is un-trustable. This is why usually 100% of spheroids is assumed.

Thank you for this comment. We have added a corresponding statement to the text: "*In fact, spheroid fractions were never found to exceed 40% when using the traditional 3+2 input regardless of the dust content in the mixed pollution plumes. A similar spheroid fraction of on average 35% has previously been reported for the inversion of 3+2 data sets obtained for Saharan dust* (Veselovskii et al., 2010). *The inverse problem for the 3+2 input data set is strongly under-determined, and thus, leads to untrustworthy spheroid fractions. This is why the inversion of 3+2 data has only been done for pure-dust conditions with the spheroid fraction set to 100%* (Veselovskii et al., 2010)."

p6, ln9: "The high values of \delta_532 lead to a dust fraction above 80% " Here the table with predicted depolarizations would be useful.

We believe that such a table could be misleading as it would require the use of suitable size distributions in the light-scattering calculations. However, these are not known a priory and could change from case to case leading to further ambiguity. We therefore decided against the Referee's suggestion. Please also see our replies to the first two comments.

What discrepancy did you obtain in retrievals?

We have now added the mean number of averaged solutions for the different input data sets together with the mean, median, and range of discrepancies related to the corresponding number of solutions to Table 2. The text has also been revised as outlined in the reply to the next comment.

Fig. 2: Why not to show retrieved real and imaginary parts? Error bars should be added.

We have now added profiles of the real and imaginary parts to the case studies in Figures 3 (former Figure 2) and 5 (former Figure 5) as well as to the correlation plots in Figures 8 and 9 (former Figures 7 and 8). In addition, we have added a new Figure 10 that presents the connection between the real and imaginary part of the refractive index and the dust ratio. We have refrained from adding error bars to the figures as they make it much harder to extract information from the figures. An example of what Figure 3 would look like with error bars for just two data sets (i.e. no depolarization input and triple-depolarization input) is shown below. However, we have added the median errors for the retrieved parameters for the respective input data sets to Table 2. A description of how these error bars have been derived has been added to Section 3.2: "*In the inversion algorithm, the obtained microphysical properties are used to re-calculate the input parameter and to assess the discrepancy between the original input data and the optical data set that would be obtained from the retrieved microphysical properties. In the analysis of the inversion calculations, we have averaged those 140 to 200 solutions (median value of 160 for the different input data sets) that revealed the smallest discrepancy to the optical input data. The mean and median discrepancies found from this approach for the different input data sets are shown in Table 2 together with the range of derived values. In general, median discrepancy increased with increasing number of input data from 1 (no depolarisation input) over 7-11 (one depolarisation input) and 13-17 (two depolarisation inputs) to 22 (three depolarisation inputs). The error of the respective parameters have been obtained as one standard deviation of averaging over the number of accepted solutions. The absolute errors presented in Table 2 refer to the median of all the error values for a respective input data set.*"

[Figure]

**Figure 3:** Inversion results of (a) effective radius, (b) 532-nm SSA, (c) spheroid fraction, and (d) volume concentration, and (e) real and (f) imaginary part of the refractive index for eight inversion runs with varying depolarisation-ratio input (colours, see also Table 2) using the input data presented in Figure 2. The dashed line in the plot of the non-spherical fraction refers to the contribution of dust to the 532-nm backscatter coefficient that can be obtained according to the procedure described by Tesche et al. (2009b). Error bars have been for results obtained from the inversion of the data sets without and with triple depolarization input, respectively.

p11, ln18: "The most realistic non-spherical fractions is found when using depolarization information at 355 nm." Have you any explanation for this?

At the moments, we have no explanation for this. It seems particularly surprising in light of the fact that depolarisation ratios inferred at small wavelengths using the Dubovik model show the largest discrepancy compared to lidar measurements (see new Figure 1).

**Anonymous Referee #3**

This work is valuable since it discusses the most useful depolarization input for retrieving dust properties from lidar measurements using the spheroid model of Dubovik et al. (2006). The Dubovik model was generated for retrieving dust microphysical properties with passive remote sensing, but it has been extensively used for dust retrievals with lidars. A discussion relevant on the Dubovik model capabilities and limitations for dust retrievals using lidar measurements is missing from the literature. Although the work in discussion does not extensively do this, it opens a discussion for the most useful input in terms of depolarization at 355, 532 and 1064 nm.

We have added a new Section 2: *Spectral δ for mineral dust from measurements and modelling* and a new Figure 1 to provide a better overview over field and lab measurements of spectral particle linear depolarisation ratios as well as findings from modelling using spheroids and irregularly shaped particles. The purpose of this now section is also to put the results presented in our study into a broader context of dust remote sensing.

New Figure 1 looks like this:

[revised manuscript text omitted]

Unfortunately the paper is very confusing since the authors present their results as if they consider a universal spheroid model. The spheroid model of Dubovik et al. (2006) cannot be considered as a universal spheroid model, due to the assumption used for the aspect ratio distribution and the mixture of prolate and oblate spheroids. This is discussed in Dubovik et al. (2006), showing that the values of P22/P11 (i.e. the phase matrix elements relevant to depolarization) can be different for different aspect ratio distributions (see Fig. 12 in Dubovik et al. (2006); the values at 180 degrees are not shown but from the scattering angles up to~170 degrees the tendency is clear). Dubovik et al. state in page 15 of their paper that: "Figure 12 illustrates the phase matrices simulated for desert dust aerosol using an ensemble of spheroids differing only by their axis ratio distributions. .... Some differences can be seen for ... P22/P11, however they are likely to be insignificant for passive remote sensing applications."

We would like to thank the Referee for these very constructive comments. Throughout the manuscript we are now emphasising that a specific version of spheroid model is being applied in our study. We hope that we can avoid the impression of providing too general conclusions this way.

This is most probably the reason the results of the work in discussion are different from results shown in Gasteiger and Freudenthaler (2014), who assumed the spheroidal shape for non-spherical particles, but assumed different aspect ratios and mixture of prolate and oblate particles, than the Dubovik model.

We have revised the discussion of the differences between our findings and the ones of *Gasteiger and Freudenthaler* (2014) for a better emphasis of the differences between the applied models. The revised text in the discussion section now reads: "*The results we obtain from our study are somewhat contradictory to the findings of* Gasteiger and Freudenthaler (2014) *and* Shin et al. (2018) *who attribute the greatest informational value and representativeness to observations of* $\delta_{1064}$. *This is likely due to the difference in the setup of the different studies. On the one hand,* Gasteiger and Freudenthaler (2014) *applied a spheroid model that was constrained in neither the aspect ratio not the oblate-to-prolate ratio of the spheroids. On the other hand,* Shin et al. (2018) *are referring to AERONET data which have been derived for scattering angles smaller than 180° (and extrapolated to the backscatter direction) and represent values for the entire atmospheric column. However, the strong and weak effects of using* $\delta_{355}$ *and* $\delta_{1064}$ , *respectively, in the inversion of lidar measurements of dust-containing aerosol layers based on the spheroid model of* Dubovik et al. (2006) *also indicate that the model's constraints on the aspect-ratio distribution and the ratio of oblates*

*to prolates have a strong effect on making full use of the informational content provided at different wavelenths in lidar applications. We stress again that the conclusions of this study are valid only for the inversion of lidar data that resorts to describe the light-scattering properties of non-spherical dust particles by means of the spheroid model of* Dubovik et al. (2006) *and that any finding might be strongly related to the weaknesses of this particular light-scattering model* (Müller et al., 2010, 2013). *Nevertheless, no other model has been applied as widely in the inversion of lidar data* (Veselovskii et al., 2010; Di Girolamo et al., 2012; Papayannis et al., 2012; Müller et al., 2013). *In addition, there has so far been no systematic investigation of the usefulness of different depolarisation input for this particular inversion setup. We therefore believe that this work will contribute to a better understanding of the usefulness and limitations of the model of* Dubovik et al. (2006) *in lidar applications as well as to further emphasise the need for a more general model to describe light-scattering by non-spherical particles at very large scattering angles.*

*Following on the initial work of* Veselovskii et al. (2010), *we have performed the first systematic investigation of the effect of all possible combinations of depolarization-related inversion input at the wavelengths of 355, 532, and 1064nm on the retrieved aerosol microphysical properties. The aim of this study is to assess the performance of the inversion procedure to lidar measurements conducted in the presence of mixture of spherical and non-spherical particles. So far, inversions on the presence of mineral dust have only been attempted if pure-dust conditions could be assumed* (Veselovskii et al., 2010; Di Girolamo et al., 2012) *or of the contribution of the non-spherical scatterers had been screened from the optical input data* (Tesche et al., 2011a, b)."

In conclusion, although the work in discussion can be a valuable tool for dust lidar retrievals, it should undergo major revisions before it is published in AMT. Specifically, the authors should shift its objective to discuss what their results are actually about, starting with changing the title to something similar to: "What are the most useful de-polarization input for retrieving the microphysical properties of dust particles from lidar measurements using the spheroid model of Dubovik et al. (2006)". Please revise the whole manuscript accordingly.

The title has been changed as suggested. We have also added comments throughout the text that we are not using a universal spheroid model but the rather specific version of *Dubovik et al.* (2006).

Last, please note that the focus of the work is on dust particles and not on "non-spherical particles" in general. Moreover

(a) change the retrieved "non-spherical fraction" to "spheroid fraction" throughout the document,

The suggested changes have been made in the text and the figures.

(b) include plots of the retrieved real and imaginary part of the refractive index in the results section and

The suggested plots of the real and imaginary part of the refractive index have been added to Figures 3, 5, 8, and 9 (former Figures 2, 4, 7, and 8). In addition, we have added a new Figure 10 that shows the connection between the real and imaginary part of the refractive index and the dust ratio.

The description of Figure 10 is "*The dependence of the retrieved real and imaginary parts of the refractive index on the dust ratio is shown in Figure 10. The upper plot shows that the retrieved real parts cover a large range of values for dust ratios smaller than 60% with data sets that include $\delta_{355}$ generally giving lower values (3+2+3 gives the lowest values) than data sets that exclude $\delta_{355}$. The range of results narrows for larger dust ratios for all inversion inputs that include depolarisation information. Only the traditional 3+2 data set gives values smaller than 1.50 for dust ratios larger than 60%. A reversed behaviour is found for the imaginary part of the refractive index in Figure 10b. Data sets that include $\delta_{355}$ generally give larger values than those that exclude $\delta_{355}$ with the traditional 3+2 data set leading to by far the highest values. This holds particularly for dust ratios larger than 60% for which the range of results from all other input data sets narrows in the same way as found for the real part.*"

[Figure]

**Figure 1:** Connection between the retrieved (a) real and (b) imaginary parts of the refractive index and the dust ratio for the different input data sets.

We have also revised the discussion to account for the findings for the refractive index in more detail: "*The inversion assumes a spectrally independent complex refractive index. In contrast, mineral dust is known to show a strong increase in the imaginary part of the refractive index with smaller wavelengths. This issue has been explored by* Veselovskii et al. (2010) *who conclude that (i) the error of the volume concentration is estimated as 17% to 25% depending on the contribution of large particles and (ii) the spectrally independent imaginary part refers to the mean value of the spectrally dependent imaginary part. A detailed investigation of the assumption of spectrally independent refractive indices is beyond the scope of this study.*"

 (c) provide a more extensive discussion on the effect of the assumption of spectrally-independent refractive index used in the analysis.

The same point has been made by Referee #2. We have added the following statement to Section 3.2 to clarify the setup of the inversion algorithm with respect to the refractive index:

"*The inversion uses a single refractive index that is independent of particle size and wavelength.*"

Regarding the effect of this constraint, we have added the following paragraph top the discussion of the findings: "*
[revised manuscript text omitted]

| SSA | 0.036 | 0.034 | 0.023 | 0.027 | 0.034 | 0.035 | 0.029 | 0.038 |
| SF | 24 | 5 | 6 | 11 | 5 | 6 | 6 | 6 |
| $v$ | 4 | 4 | 3 | 5 | 4 | 6 | 4 | 6 |
| $m_{\text{r}}$ | 0.068 | 0.059 | 0.046 | 0.059 | 0.065 | 0.070 | 0.062 | 0.071 |
| $m_{\text{i}}$ | 0.006 | 0.005 | 0.003 | 0.004 | 0.005 | 0.005 | 0.004 | 0.005 |
| **linear fits, Figure 7** | | | | | | | | |
| Slope | 0.13 | 0.87 | 0.41 | 0.09 | 0.94 | 0.90 | 0.42 | 0.95 |
| Intercept | 18 | 8 | 58 | 80 | 9 | 11 | 62 | 13 |
| $R^2$ | 0.17 | 0.69 | 0.33 | 0.03 | 0.73 | 0.68 | 0.46 | 0.73 |

---

## Author Response (AR2)

We would like to thank the Editor for his comments and suggestions. Please find below our replies and changes to the manuscript marked in blue.

The paper concludes that δ355 provides sufficient information for a 3+2+X dust retrieval when using the Dubovik model, but no explanation is provided on why this happens. Gasteiger et al. (2011) discuss the possibility that for larger size parameters δ is similar for spheroids and irregularly-shaped particles. This may be a possible explanation: δ532 and δ1064 refer to smaller size parameters than δ355, thus their values for spheroid particles are not similar to their values for irregular-shaped (more realistic) particles. This may be the reason that δ532 and δ1064 do not provide good retrievals when the spheroid model (of Dubovik et al. (2006)) is used, whereas δ355 does. Please include this discussion in the paper.

We would like to thank the Editor for this valuable addition to the discussion of our findings. We have added the following statement to the discussion section: *"Furthermore, Gasteiger et al (2011) have discussed the possibility that δ becomes similar for spheroids and irregularly-shaped particles for larger size parameters. For the larger size parameters equivalent to measurements at 355 nm, it might therefore be that $\delta_{355}$ as obtained for spheroid particles is similar to that of more realistic, irregular-shaped particles. As $\delta_{532}$ and $\delta_{1064}$ refer to smaller size parameters, it would consequently be more likely that spheroids fail to properly describe light-scattering properties at these wavelengths."*

Page 4, lines 13-14, "i.e. δ=0 for spherical particles (or those that appear spherical with respect to the considered wavelength) and increases with particle non-sphericity": This is not always true. As shown in Mishchenko and Hovenier (1995) there is no systematic dependence of particle asphericity with the light depolarization, with cases presented therein of strongly aspherical particles producing depolarization values close to zero. Please rephrase accordingly.

Thank you for this comment. We have refined the statement to: *"…increases with particle non-sphericity for the particle shapes commonly found for atmospheric aerosols and ice crystals."*

Page 5, lines 26-27, "In both cases… below 0.01": This is the conclusion of Mishchenko et al. (2016) but not of Gialitaki et al. (2019). Please rephrase accordingly.

The statement has been revised to: *"In the studies of Bi et al. (2018) and Mishchenko et al. (2016), the model particles…"*

Page 8, lines 6-9, "For the measurements… (Set I)": This paragraph is repeated in lines 26-29.
We have deleted the repetition of this paragraph.

Page 10, lines 17-18, "This indicates… the retrieval.": I think you should provide more info/examples to justify this statement, or avoid making it.
The statement has been removed from the manuscript.

Page 11, lines 33-34, "(ii) the data sets… to the obtained dust ratio,": Fig. 7a shows that the input of δ355+δ532+δ1064 and of δ355+δ1064 provide good comparisons with the dust ratio. Please rephrase.

The Editor is correct. The statement has been changed to *"the data sets without $\delta_{355}$"*

References
Mishchenko, M. I., and Hovenier, J. W.: Depolarization of light backscattered by randomly oriented nonspherical particles. Opt. Lett. 20, 1356–1358, 1995.